# Cell cycle dependent coordination of surface layer biogenesis in *Caulobacter crescentus*

Matthew Herdman [1], Buse Isbilir [2], Andriko von Kügelgen [1,2], Ulrike Schulze[3], Alan Wainman [1] & Tanmay A. M. Bharat [2] ✉

Surface layers (S-layers) are proteinaceous, two-dimensional paracrystalline arrays that constitute a major component of the cell envelope in many prokaryotic species. In this study, we investigated S-layer biogenesis in the bacterial model organism *Caulobacter crescentus*. Fluorescence microscopy revealed localised incorporation of new S-layer at the poles and mid-cell, consistent with regions of cell growth in the cell cycle. Light microscopy and electron cryotomography investigations of drug-treated bacteria revealed that localised S-layer insertion is retained when cell division is inhibited, but is disrupted upon dysregulation of MreB or lipopolysaccharide. We further uncovered that S-layer biogenesis follows new peptidoglycan synthesis and localises to regions of high cell wall turnover. Finally, correlated cryo-light microscopy and electron cryotomographic analysis of regions of S-layer insertion showed the presence of discontinuities in the hexagonal S-layer lattice, contrasting with other S-layers completed by defined symmetric defects. Our findings present insights into how *C. crescentus* cells form an ordered S-layer on their surface in coordination with the biogenesis of other cell envelope components.

Cell envelopes of prokaryotes are complex, multi-layered structures that fulfil a variety of roles, such as mediating interactions with the environment including neighbouring cells, regulating import and export of material, and protection from external attack[1,2]. Many prokaryotes including archaea, diderm, and monoderm bacteria express a macromolecular, proteinaceous sheath known as the surface layer (S-layer) as one of the most exterior part of their cell envelope[3–6]. There is increasing evidence suggesting that S-layers are prevalent in prokaryotes, with a majority of bacteria and most archaea expressing an S-layer on their envelopes[4,7]. S-layers are two-dimensional arrays made up of repeating copies of S-layer proteins (SLPs). Since SLPs are the highest copy number proteins in many prokaryotic cells, by several estimations they are one of the most abundant protein family found in nature[3,8,9]. Given their position as one of the outermost components of the envelope, coating the entire cell surface, it is no surprise that S-layers are important for several aspects of cell biology and are suggested to be an ancient form of a cellular exoskeleton[10,11]. S-layers have

been implicated in the maintenance of cell size and shape[11,12], evasion from predators[13], attachment to substrates[14,15], and as a protection against a range of environmental pressures[16–18].

The understanding of the evolution of S-layers is far from complete, since SLPs appear in even the most deep branching lineages of prokaryotes and show a high level of sequence variability[4,12,19,20]. Despite this variability, S-layers share several organisational features; for example, SLPs are often bipartite in nature, encoding distinct lattice-forming and cell-anchoring domains, often within the same protein[4,21–23]. Secondly, many SLPs utilise metal ions to facilitate both their retention on the cell surface, as well as lattice assembly[24–31]. Thirdly and intriguingly, insertion of new S-layer subunits is localised at the mid-cell and cell poles in many prokaryotes, including archaeal[32], Gram-positive[33] and Gram-negative bacterial species[4,32,34].

One of the best characterised systems for studying bacterial S-layers is provided by the model organism *Caulobacter crescentus*[35]. The S-layer of *C. crescentus* is comprised of a single SLP

[1]Sir William Dunn School of Pathology, University of Oxford, Oxford OX1 3RE, UK. [2]Structural Studies Division, MRC Laboratory of Molecular Biology, Cambridge CB2 0QH, UK. [3]Cell Biology Division, MRC Laboratory of Molecular Biology, Cambridge CB2 0QH, UK. ✉e-mail: tbharat@mrc-lmb.cam.ac.uk

called RsaA[36], which has the prototypical bipartite arrangement of SLPs[4]. We have reported the X-ray structure of the C-terminal domain of RsaA (RsaA$_{CTD}$), consisting of residues 250–1026, which form the highly-interconnected outer S-layer lattice[29], and solved the cryo-EM structure of the N-terminal domain (RsaA$_{NTD}$, residues 1–249) in complex with the O-antigen of lipopolysaccharide (LPS)[28,37], on which the S-layer is anchored[38]. S-layers of multiple species assemble in a metal-ion dependent manner[24,26,27,30,39]. Likewise, *C. crescentus* requires a high concentration of extracellular calcium ions for SLP oligomerisation and retention on the cell surface[31,40]. Further, new S-layer insertion in *C. crescentus* is localised at the mid-cell and cell poles, by a mechanism that is not yet understood[31,34]. In general, S-layer-expressing prokaryotes synthesise SLPs at high levels, and in the case of *C. crescentus*, RsaA has been suggested to account for between 10 and 31% of total protein content of the cell[41,42] and its concentration appears to be tightly regulated to prevent cytoplasmic build-up of excess protein[41–44]. Given the material and energetic demand imposed on the cell by S-layer, it is reasonable to expect that S-layer assembly is also carefully regulated during the cell cycle.

In this study, we have investigated the cell cycle dependence of S-layer biogenesis in *C. crescentus*, using cellular fluorescence microscopy and electron cryotomography (cryo-ET), which allowed us to visualise the S-layer arrangement on cells, down to the level of single S-layer hexamers. Our results show that S-layer biogenesis is tightly linked with cell elongation and cell growth. We provide evidence showing that cell division and cell envelope biogenesis are regulated at multiple levels, providing new insight into the exciting field of S-layer biology, offering clues to why all S-layers (thus far) appear to be inserted at discrete locations in the cell.

## Results

### S-layer insertion is localised to regions of cell-cycle dependent envelope growth in *C. crescentus*

To understand the cell-cycle dependency of S-layer insertion, we utilised a dual-labelling approach previously described in our study of calcium binding by RsaA[31,45]. Briefly, to distinguish between old and newly inserted regions of the S-layer, we pulse-saturated the surface of *C. crescentus* cells expressing RsaA-467-SpyTag (RsaA-467-ST) with SpyCatcher-mRFP1 (SC-mRFP1), followed by washing and chase labelling with SpyCatcher-sfGFP (SC-sfGFP) during exponential growth (Methods). Following labelling of the surface available SpyTags, we observed *C. crescentus* cells with distinct fluorescent regions of mRFP1- and sfGFP-labelling, corresponding to the pulse and chase respectively (Fig. 1a). Cell populations were asynchronous and fluorescence profiles varied depending on the cell size and cell cycle stage. Non-dividing cells showed limited or polar sfGFP labelling (Fig. 1b), while pre-divisional cells had a strong mid-cell sfGFP signal (Fig. 1c), in agreement with previous reports[31,34].

To determine the relationship between S-layer localisation and cell size, cell profiles were ordered according to cell length in MicrobeJ using previously described methods[46], and their fluorescent signals were then visually inspected (Fig. 1d, e). This analysis revealed a clear temporal progression of new S-layer biogenesis (labelled with sfGFP signal) from poles to mid-cell. To further quantify the relationship between the cell cycle stage and the labelling pattern, the fluorescence profiles of non-dividing swarmer cells (cell length <2 μm) and dividing cells (assigned by the presence of a mid-cell invagination) were normalised and plotted against relative cell length. As expected from the visual inspection of the data, non-dividing cells (Fig. 1f) showed a stronger normalised sfGFP signal at their poles, while dividing cells had

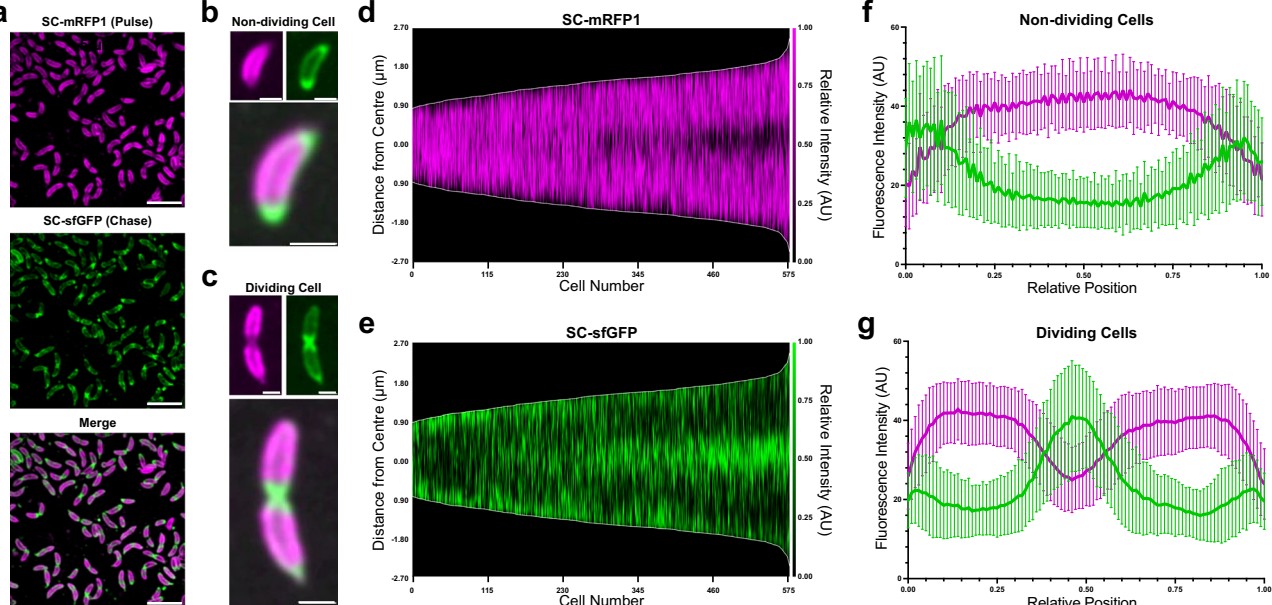

**Fig. 1 | Incorporation of RsaA into the growing S-layer of *C. crescentus* relocates from the cell poles to the mid-cell during cell development. a** Micrographs of *C. crescentus* RsaA-467-ST cells pulse-chase labelled using SC-mRFP1 (top, magenta), SC-sfGFP (middle, green). Merged channels (bottom) show distinct localisation of the two SC-conjugates along the cell surface. Micrographs were gaussian filtered to remove noise. Scale bar = 5 μm. **b, c** Micrographs (SC-mRFP1 (top left, magenta), SC-sfGFP (top right, green), and merged channels (bottom)) of a representative non-dividing and dividing *C. crescentus* cell. Polar and mid-cell localisation of newly inserted S-layer being more prominent in non-dividing and dividing cell populations, respectively. Scale bar = 1 μm. The data presented is representative of one experiment, repeated three times with comparable results. **d, e** Demograph showing normalised fluorescent profiles of dual-labelled *C. crescentus* cells (n = 575 cells), ordered by ascending length. **d** SC-mRFP1 signal corresponds to old S-layer, while (**e**) SC-sfGFP signal represents new S-layer. Shorter, non-dividing cells show a propensity for polar localisation of new S-layer, while longer cells show mid-cell localisation of the sfGFP signal. Relative intensity profiles of mRFP1 and sfGFP in (**f**) non-dividing cells and (**g**) dividing *C. crescentus* cells (n = 100 cells for both plots). Points were selected across the medial axis of each cell, and the normalised signal plotted by relative position along the cell. The thick green and magenta lines show mean value and error bars denote standard deviation.

a much stronger signal at their mid-cell (Fig. 1g). The LPS-bound S-layer has been previously shown to be non-diffusing relative to the underlying cell following oligomerisation[34]. To confirm this, we labelled the cell surface with SC-mRFP1, and imaged the cells live, during growth at 30 °C. As expected, we observed the same localisation of the old, SC-mRFP1-labelled S-layer on the cell body, away from the mid cell and cell poles (Supplementary Fig. 1). As the cells grew, these regions remained stationary and non-diffusing relative to the cell, evident in kymographs of growing cells. The non-fluorescent areas in the kymographs, which appeared to expand, represent those regions that would have been occupied by SC-sfGFP if we had dual-labelled the cell surface, i.e., the new S-layer, shown above in our pulse-chase experiments (Fig. 1).

As a control, *C. crescentus* cell cultures were briefly synchronised using density centrifugation with Percoll and pulse-chase labelled as above, resulting in a similar labelling pattern to non-synchronised cells (Fig. 2). In addition, cells labelled in this manner often displayed variable stalk labelling, including occasional dual-labelled stalks (Fig. 2b). Stalks are polar appendages produced by division-competent *C. crescentus* cells, which are also encompassed by an S-layer[29,36]. Stalk biogenesis is driven by the recruitment of peptidoglycan (PG) remodelling machinery at the old cell pole[47–49], following a rapid transition from swarmer to sessile cell type in *C. crescentus*, the latter representing the dividing population[49–51]. In dual-labelled cells, new S-layer labelled in SC-sfGFP was localised toward to stalk base, further affirming the potential colocalisation of S-layer biogenesis with underlying cell wall turnover.

## Inhibition of cell-division does not cause delocalisation of S-layer insertion

To further explore processes underpinning the localisation of new RsaA insertion into the S-layer, we sought to perturb the cell cycle of *C. crescentus* using compounds that interfere with various aspects of the cell cycle, to assess the local accumulation of new and old S-layer material in response to the treatment. To aid comparison between treatments, the distinctive regions of old (mRFP1) and new (sfGFP) S-layer observed using our pulse-chase method were quantified for co-localisation using Pearson's Correlation Coefficient (PCC)[52,53]. PCC quantifies the linear correlation between two datasets (in this case, two channels of a fluorescence image): a PCC > 0 suggests there is co-localisation between the two channels, whereas PCC < 0 shows anticorrelation[52,54–56]. As expected from visual inspection of our data, in the absence of cell-cycle perturbing compounds, pulse-chase labelled *C. crescentus* cells exhibited a strong anticorrelation (average PCC = −0.46) between the old and the new S-layer regions (Fig. 3a, d).

To test how S-layer biogenesis may be affected by disrupted cell division, we next treated cells with cephalexin, a cephalosporin antibiotic that inhibits cell division by inhibiting penicillin-binding protein 3 (PBP3), which is a divisome-associated protein mediating PG synthesis at the mid-cell during division[57–59]. We used a concentration of cephalexin that allowed the cells to grow, thus remaining amenable to our pulse-chase labelling methods, but with partial disruption of cell division. Based on growth curves using varying concentrations of cephalexin (Supplementary Fig. 2a) and quantification of cephalexin-induced cell elongation (Supplementary Fig. 3), we supplemented the PYE growth medium with 50 μg/mL cephalexin. This sub-lethal level of cephalexin exposure significantly inhibited cell division, and resulted in the formation of lines of connected filamentous cells (Supplementary Fig. 3), consistent with previously published work[60]. These filamentous cells were labelled in the same manner as untreated cells, but with a prolonged chase (3 h) to allow for growth. Despite cephalexin's marked impact on cell morphology, which could be caused by inhibition of other PBPs beyond PBP3, remarkably cephalexin-treated cells retained a dual-labelled S-layer pattern with distinct regions of old and new S-layer (Fig. 3b and Supplementary Fig. 4). Repeating the co-localisation analysis in these cells confirmed that the old and new S-layer regions were strongly anticorrelated, almost to the same extent as untreated cells (Fig. 3d, average PCC = −0.45, not significantly different from untreated, as measured by a two-tailed Student's *t* test). The localisation pattern is

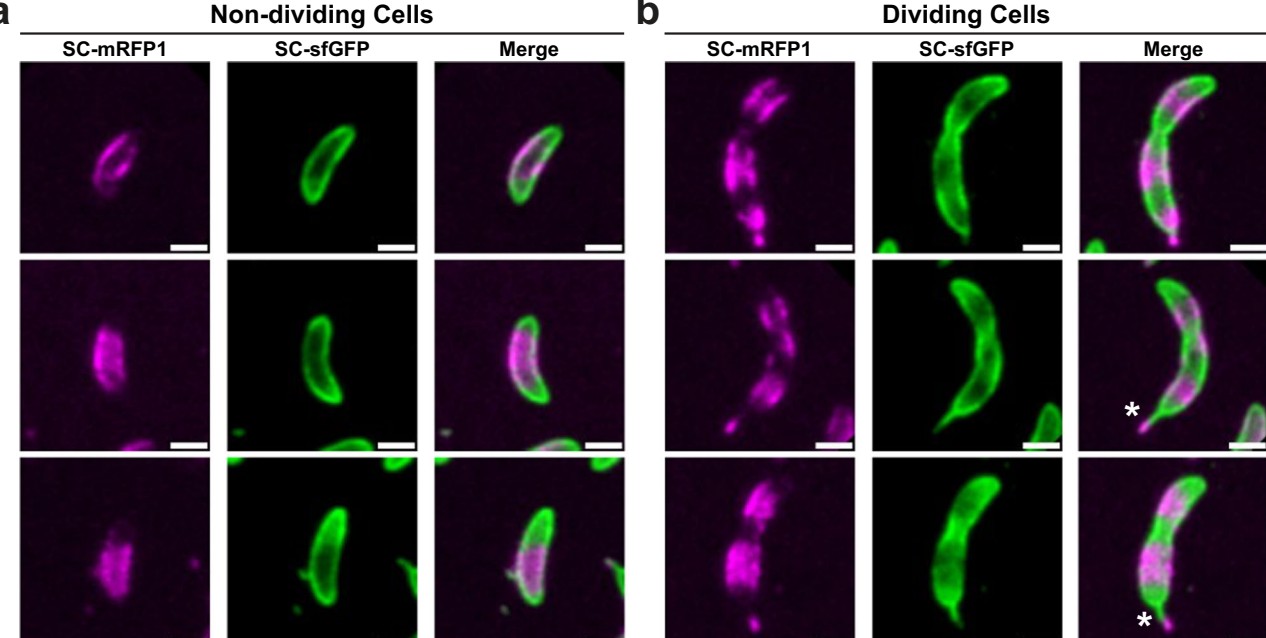

**Fig. 2 | S-layer localisation patterns in dividing and non-dividing *C. crescentus* cells.** Comparison of labelling in (**a**) non-dividing cells and (**b**) dividing, stalked cells. Cells were synchronised prior to pulse-chase labelling using SC-mRFP1 and SC-sfGFP as described. The micrographs are taken from a single experimental replicate, of which three were performed with comparable results. Polar labelling can be seen in all cells, but mid-cell labelling is only apparent in dividing cells. Additionally, dual-coloured stalks (SC-sfGFP at the base of the stalk, and SC-mRFP1 at the stalk tip) are indicated by an asterisk. This is consistent with previous research that shows that new stalk material is created from the base of the stalk[120]. Scale bars = 1 μm.

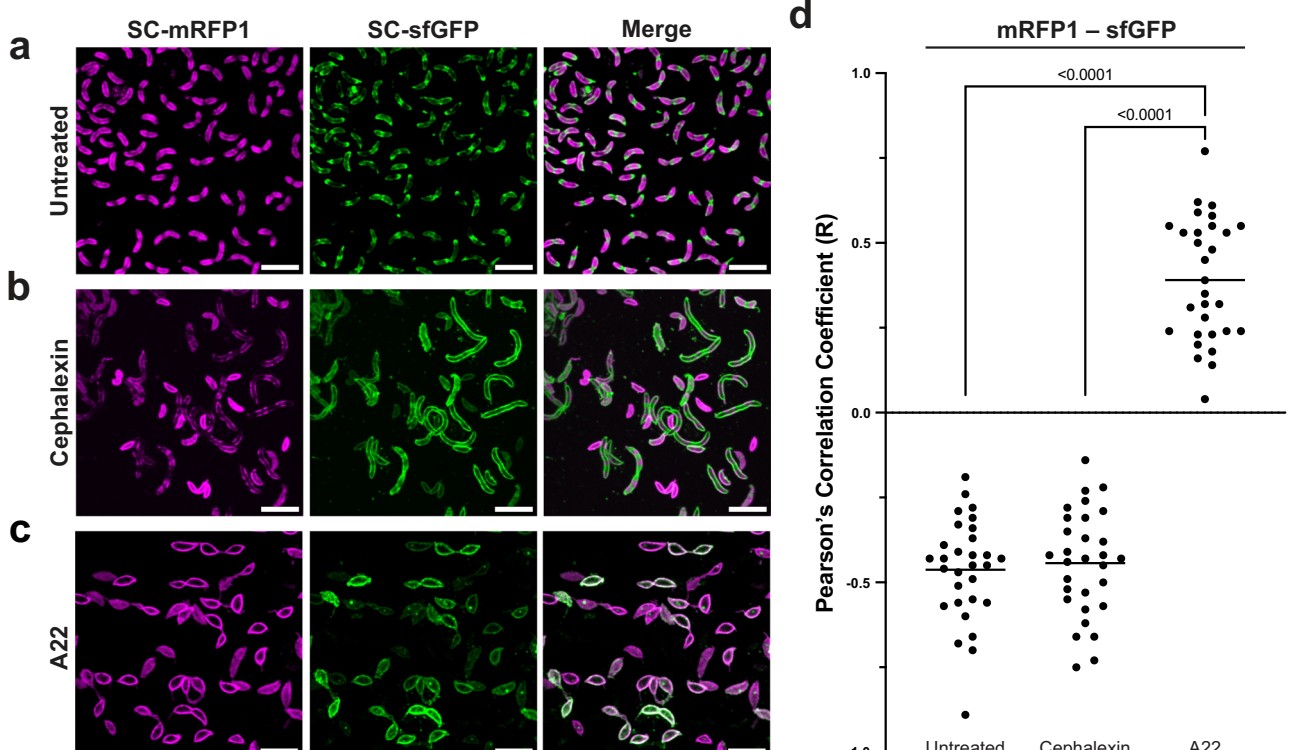

**Fig. 3 | Inhibition of MreB, but not cell division, results in delocalisation of S-layer proliferation in *C. crescentus*. a–c** Micrographs of dual-labelled *C. crescentus* RsaA-467-ST cells. SC-mRFP1 (left, magenta), SC-sfGFP (centre, green) and merged channels (right). Cells were pulse-chase labelled using the same procedure to that of Fig. 1, but under varying conditions, including (**a**) no treatment, (**b**) cells treated with 50 μg/mL cephalexin, and (**c**) 3 μg/mL A22. Scale bars = 5 μm. Each labelling experiment was repeated at least three times with comparable results. The micrographs presented are representative of a single experimental repeat.
**d** Analysis of colocalisation by PCC show that untreated and cephalexin treated cells have a negative colocalisation (R value) between the mRFP1 and sfGFP

channels, as expected given the visually observable separation of old and new S-layer (horizontal line shows the mean). A22 treated cells show a positive R value and colocalisation between the both channels, suggesting MreB inhibition has resulted in the loss of discrete localisation of RsaA insertion into the S-layer during cell growth. Ordinary one-way ANOVA analysis of the data shows a strong significant difference between A22 treated cells and both other conditions ($p < 0.0001$), whereas comparing untreated and cephalexin treated cells showed no significant differences (two-tailed Student's *t* test, $p = 0.625$). $n = 30$ cells analysed for each treatment condition. Source data are provided as an accompanying Source Data file.

also consistent between untreated and cephalexin-treated cells, with new S-layer localising to the mid-cell and cell poles (Supplementary Fig. 4a–d). The cephalexin-treated cells often showed some additional new S-layer in the cell body, equidistant between the cell poles and mid-cell, at potential sites of disrupted cell division.

While S-layer localisation had clearly been retained, we sought to confirm the S-layer in division-disrupted, cephalexin-treated cells had a normal appearance. Cells were vitrified on electron microscopy (EM) grids following treatment with 50 μg/mL cephalexin and imaged using cryo-ET. Reconstructed tomograms confirmed that the S-layer is positioned ~18 nm away from the outer membrane, forming a hexagonal lattice (Fig. 4, Supplementary Fig. 5 and Supplementary Movie 1), consistent with tomograms of untreated cells in published data[31]. These results together suggest that disrupting cell division via cephalexin treatment does not affect the localisation of S-layer assembly, nor does it disrupt RsaA secretion.

To ensure the observed retention of discrete S-layer localisation is not concentration-dependent, cells were incubated with an increased concentration of cephalexin (100 μg/mL), which results in arrest of culture growth, as measured by $OD_{600}$ (Supplementary Fig. 2a). In addition, we incorporated the division-disrupting drug mitomycin-C (MMC) into the labelling protocol at lethal concentrations in an additional experiment (Supplementary Fig. 2b). MMC is a well-characterised DNA-crosslinking agent[61] that induces cell filamentation via the SOS response[62]. Under MMC treatment, cells inhibit their division, preventing the propagation of mutations

to progeny cells. This has been well characterised in *C. crescentus*[63,64] and other bacterial species, including *Streptomyces venezuelae*[65,66]. As expected, both the increased concentration of cephalexin and MMC resulted in cell elongation (Fig. 5a). In addition, both conditions produced cells with an anticorrelated localisation of old and new S-layer, evidenced by strong, negative PCC R-values, comparable to untreated cells (Fig. 5b). Together, this confirms that disrupting cell division through multiple pathways does not lead to delocalisation of S-layer biogenesis.

### Disruption of the cytoskeletal protein MreB delocalises S-layer insertion
Having shown that interrupting cell division using cephalexin or MMC did not disrupt the discrete localisation of S-layer biogenesis, we next investigated the effect of disrupting cell elongation and rod morphogenesis by the sequestering of the bacterial actin homologue MreB. MreB is a major cytoskeletal component of *C. crescentus*, mediating cell elongation, stalk biogenesis, and cell polarity[48,67,68]. To disrupt MreB, we treated *C. crescentus* cells with the compound A22, which has been shown to bind to MreB and disrupt cell shape[67,69] and cell polarity[68,70]. We investigated the effect of disruption of MreB filaments on S-layer biogenesis using sub-lethal concentrations of A22 (Supplementary Fig. 2c). Strikingly, S-layer integration at the surface of these A22-treated cells was delocalised (Fig. 3c, d). Furthermore, pulse-chased labelled cells adopted a "lemon"-shape and showed several regions of new S-layer without the previously observed mid-cell or polar

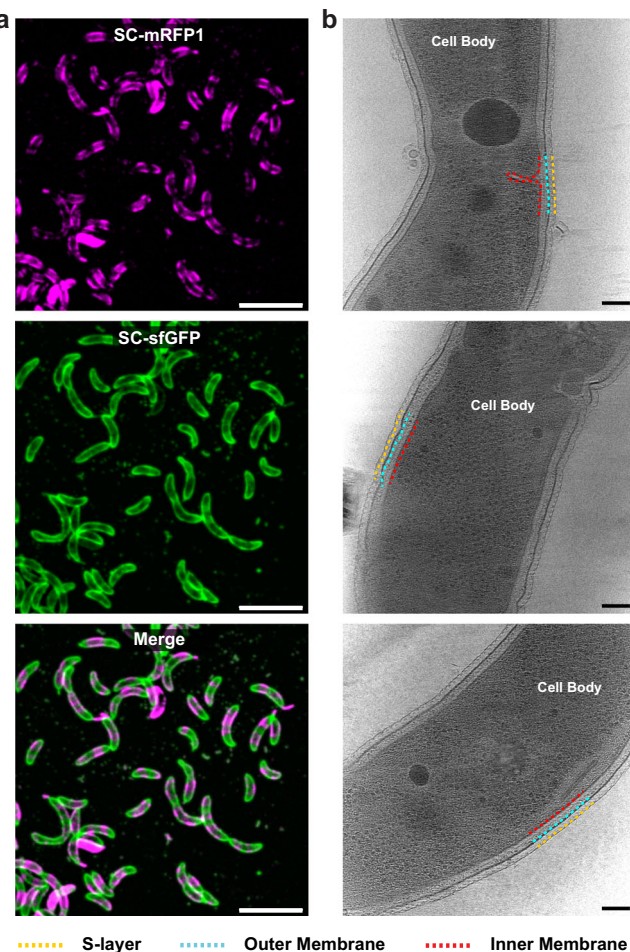

**Fig. 4 | Elongated, cephalexin-treated cells retain a continuous S-layer bound to the OM. a** Micrographs of dual-labelled, cephalexin-treated *C. crescentus* RsaA-467-ST cells, SC-mRFP1 (top, magenta), SC-sfGFP (centre, green) and merged channels (bottom). Scale bar = 5 μm. Cephalexin-treatment experiments were repeated at least three times with comparable results. The micrographs presented are representative of a single experimental repeat. **b** Slices through tomograms of vitrified cephalexin-treated *C. crescentus* cells, showing a complete S-layer bound to the OM. Scale bar = 50 nm.

localisation. Quantification for co-localisation (Fig. 3d) confirmed that this treatment led to loss of the anticorrelation between new and old S-layer, as illustrated by a positive average PCC (R-value 0.44). The PCC analysis results were significantly different to that of both untreated and cephalexin treated cells (*p* < 0.0001, measured by two-tailed Student's *t* tests). Furthermore, the patterns of new and old S-layer localisation observed in these experiments were significantly different from those seen in control untreated and cephalexin-treated cells (Supplementary Fig. 4).

Broadly, two different types of labelling patterns were observed in "lemon"-shaped A22-treated cells (Fig. 6). The first type exhibited almost no new S-layer labelling, with cells primarily stained for old S-layer. In the second type, old and new S-layers both appeared delocalised (Fig. 3c and Fig. 6). To better understand these labelling patterns and to examine ultrastructure of the S-layer in the A22-treated cells, we performed cryo-ET of A22-treated cells (Fig. 6c, d and Supplementary Fig. 5). In line with the fluorescence microscopy, tomograms also confirmed two phenotypes, both of which were "lemon"-shaped and possessed an S-layer with similar morphological parameters to that of the wild-type S-layer (Fig. 6c, d and Supplementary Movie 2). One set had severe cellular disruption including invaginated membranes (Fig. 6d, Supplementary Fig. 5 and Supplementary

Movie 3); these membrane distortions are likely responsible for the patchy fluorescence and disrupted S-layer localisation.

Dysregulation of MreB should be cautiously interpreted, as several cellular processes are potentially being simultaneously disrupted. However, the loss of discrete regions of old and new S-layer, which were observed in both untreated and cephalexin-treated cells, taken together with the retention of an ordered and intact S-layer, suggests that the effects on localised biogenesis are derived from MreB inaction.

### LPS-inhibition by polymyxin-B, but not inhibition of protein synthesis, disrupts discrete S-layer localisation

Given the dramatically different effects on S-layer localisation following exposure to cell-cycle disrupting compounds, we sought to explore how biogenesis might be affected by targeting aspects of RsaA synthesis and S-layer assembly. The addition of sublethal concentrations of polymyxin B (7.5 μg/mL, Supplementary Fig. 2d), an LPS-binding antibiotic, resulted in a dramatic loss of anticorrelation between old and new regions of S-layer, confirmed by PCC analysis (Fig. 6). While the exact mechanisms by which polymyxin B disrupts localised S-layer biogenesis cannot be inferred, our results show clearly that localised S-layer biogenesis is LPS dependent. The activity of polymyxin B is proposed to depend on its binding to lipid A[71], leading to disruption. An alternate hypothesis is that polymyxin B displaces cations like $Ca^{2+}$ from the LPS causing inhibition[72-74,40]. The presence of SC-associated fluorescence suggests that RsaA-467-ST, and therefore the O-antigen and lipid A, are all present on our polymyxin-B-treated cells. It is possible that polymyxin B binding to the LPS disrupts cation and RsaA binding, resulting in loss of the S-layer and an increased frequency of gaps, leading to the observed loss of anticorrelation of new and old S-layer regions.

In contrast, inhibition of protein synthesis by the addition of chloramphenicol (Supplementary Fig. 2e) appeared to have no observable effect on new and old S-layer anticorrelation, resulting in cells with labelling comparable to that of untreated cells (Supplementary Fig. 2f). Chloramphenicol, which acts by binding the 50S subunit of the bacterial ribosome, inhibits protein synthesis[75]. This inhibitory effect, despite RsaA being the highest copy number protein of the cell, is a nonspecific effect, resulting in a global reduction of protein synthesis. It is unsurprising then that it results in a significant growth defect (Supplementary Fig. 2e) and, while cells are slowed in their growth, they retain their usual labelling pattern of separated regions of new and old S-layers.

### Cell wall turnover precedes S-layer biogenesis at the mid-points of dividing cells

Given the known dependence of PG biogenesis on MreB in *C. crescentus*[48], we next explored the relation between new PG and new S-layer by labelling newly synthesized PG using fluorescent D-amino acids[76] alongside old and new S-layer. PG labelling with HADA (a blue fluorescent D-amino acid) showed distinct fluorescent punctae in cells (Fig. 7a), seen previously in several bacteria[76], including *C. crescentus*[77]. A visual inspection of fluorescent images suggested the co-localisation of new PG and new S-layer insertion. To test this hypothesis, we obtained profiles along the length of each cell and ordered each cell according to their lengths (Fig. 7b–d). This analysis revealed that HADA fluorescence was localised at the mid-cell in short cells in earlier stages of the cell cycle, while the integration of new S-layer material, as indicated by the presence of sfGFP, occurs in longer cells at later stages of the cell cycle, suggesting that PG turnover precedes S-layer biogenesis (Fig. 7b–d and Supplementary Fig. 6). New S-layer insertion begins at the mid-cell in dividing cells, regions where no old S-layer is detected. In the longest cells analysed, new PG insertion was not detected, despite

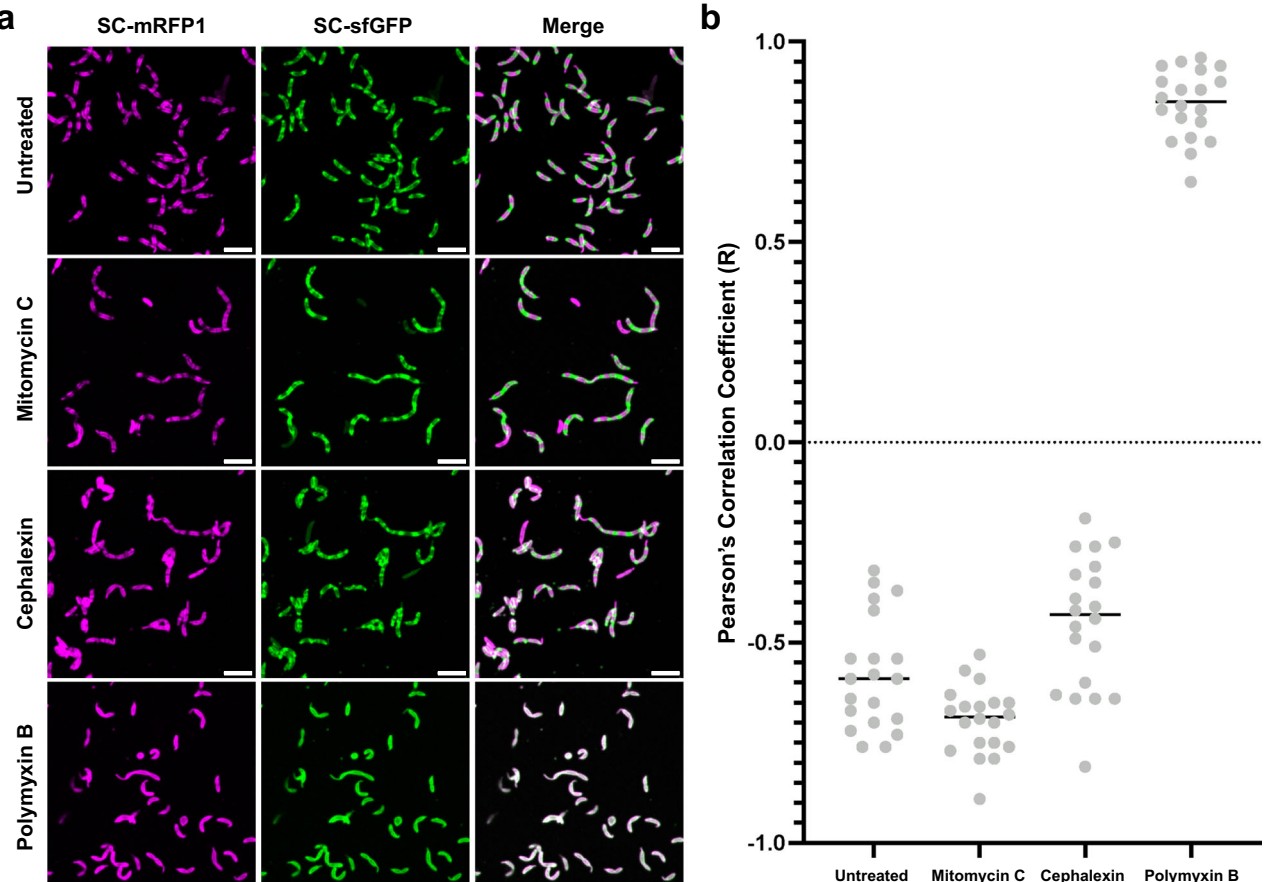

**Fig. 5 | Quantification of spatial separation of old and new S-layer following exposure to MMC, higher concentrations of cephalexin, and polymyxin B.**
**a** Pulse-chase labelled *C. crescentus* cells following growth in PYE media with no drug (untreated), or PYE supplemented with one of three compounds: 0.5 μg/mL mitomycin C (MMC); 100 μg/mL cephalexin; 7.5 μg/mL polymyxin B. Scale bar = 5 μm. **b** Co-localisation of mRFP1 and sfGFP fluorescence was quantified using Pearson's Correlation Coefficient (PCC). Untreated cells and those treated

with cell division-inhibiting compounds MMC and cephalexin showed negative PCC scores (average R-values −0.58, −0.70, −0.45, respectively). Following exposure to the LPS inhibiting peptide polymyxin B, cells showed positive PCC scores (average R-value 0.84), showing a disruption of spatially localised S-layer biogenesis. $n = 20$ cells analysed for each treatment condition. Data points (●) represent PCC results for individual cells. Central line indicates mean. Source data are provided as an accompanying Source Data file.

the presence of new S-layer insertion, indicating that PG insertion concludes around the time new S-layer insertion begins.

To confirm quantitatively that new PG and new S-layer is inserted in the same locations in cells, we repeated the co-localisation analysis described above (Fig. 3), measuring HADA fluorescence co-localisation with both new and old labelled S-layer (Fig. 7e–g). In non-dividing cells, there was no significant difference between the co-localisation measured between new PG and new or old S-layer (PCC = −0.10 new PG/new S-layer and PCC = −0.12 new PG/old S-layer). In contrast in dividing cells, new PG was co-localised with new S-layer (PCC = 0.30) rather than with old S-layer (PCC = −0.15), in a statistically significant difference (measured by a two-tailed Student's *t* test, *p* < 0.0001). These observations suggest that cell wall expansion is a driving force in cell envelope growth and a predictor of local S-layer biogenesis. The localisation of HADA fluorescent signal to the cell pole was also evident (Fig. 7d), likely corresponding to the aforementioned stalk biogenesis machinery and, potentially, recent division events. Due to the speed and unpredictable timing of stalk synthesis, as well as the temporal offset associated with labelling the separate envelope components, colocalisation of cell wall and S-layer biogenesis events at the stalk is more challenging to investigate. In several cells, the polar HADA signal does appear to colocalise with new S-layer labelled in SC-sfGFP (Fig. 7e), as expected following the observation of dual-coloured stalks (Fig. 2b).

## S-layer insertion occurs at regions of discontinuity in the S-layer lattice

Having studied the cell-cycle dependence of S-layer biogenesis, we next scrutinized how new RsaA molecules insert themselves into a pre-existing two-dimensional lattice packed with proteins spanning the cell envelope. For this, we utilised cryo-ET data of *C. crescentus* cells, focusing on the mid-cell, as done previously[78,79], or the cell poles (Fig. 8), i.e., regions of the cell where we have shown the new S-layer is inserted (Fig. 1). Cryo-ET allowed us to observe the ultrastructure of the S-layer, allowing us to go beyond the diffraction-limited optical microscopy pictures to study the morphology of the new S-layer insertion sites (Fig. 8). Unexpectedly, we observed disruptions in the S-layer lattice at the S-layer biogenesis sites (Fig. 8). As a control, we vitrified dual S-layer-labelled *C. crescentus* cells on EM grids for cryo-correlated light and electron microscopy (cryo-CLEM). Cryo-light microscopy of the vitrified cells, although limited in resolution in our widefield setup, allowed us to identify cells with clear dual labelling and identifiable sites of new S-layer insertion (Supplementary Fig. 7a, b). These cells were then located in the electron microscope by overlaying the light microscopy images with overview images of EM grid squares. Cryo-ET of these dual labelled cells confirmed disruptions in the S-layer at the site of new S-layer insertion (Supplementary Fig. 7c), confirming our cryo-ET observations (Fig. 8a–f). These regions contain points of S-layer discontinuity, where either two lattices appear to overlap, or rows of hexamers are missing (Fig. 8a–f). At

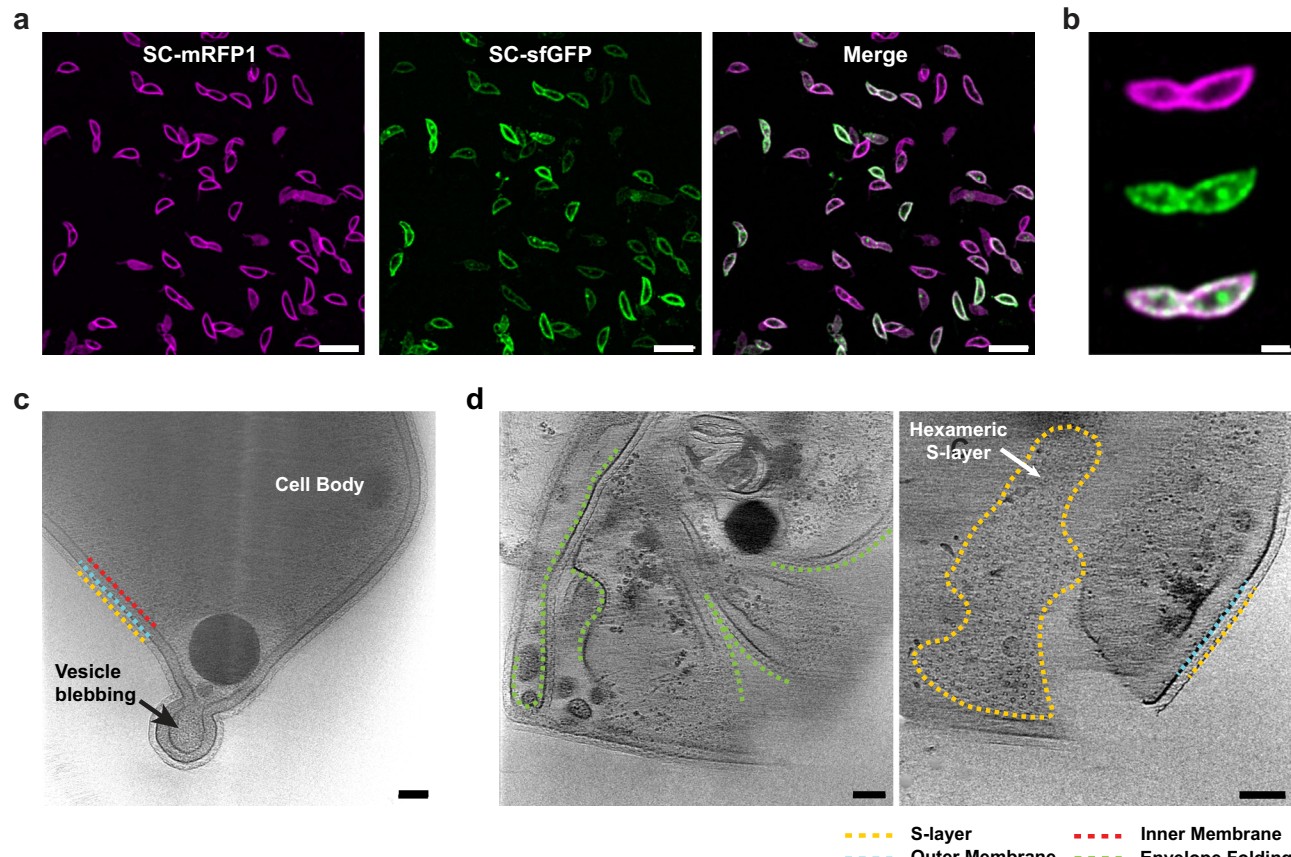

**Fig. 6 | A22-treated cells undergo dramatic envelope folding, retaining a continuous S-layer. a** Micrographs of dual-labelled, A22 treated *C. crescentus* RsaA-467-ST cells. Both channels and the merge are shown. Scale bar = 5 μm. **b** A representative A22-treated cell showing "patchy" S-layer labelling with overlapping sfGFP and mRFP1 signals. Scale bar = 1 μm. A22-treatment experiments were repeated at least three times with comparable results. The micrographs presented are representative of a single experimental repeat. **c**, **d** Slices through tomograms of vitrified, A22-treated *C. crescentus* cells. **c** Enlarged cell showing budding vesicle forming at the presumed cell pole. **d** Deformed cells showing folding of envelope resulting from A22 exposure. Folds in the envelope have been labelled (green). The second panel shows a higher Z-slice of the same reconstructed tomogram, where the top view of the S-layer is clearly visible and adopts a regular hexagonal arrangement. Scale bar = 50 nm.

several areas, two separate two-dimensional sheets of S-layers appear to meet, showing up as a line defect on the cell surface. Given their placement at regions of cell-envelope expansion, these likely represent regions of RsaA insertion or lattice formation.

## Discussion

Based on our analyses, we suggest a new model of S-layer biogenesis, which is dependent on turnover and expansion of the cell wall, leading to growth of the cell envelope (Fig. 9). We suggest that areas of cell growth contain new membranes and potentially freshly secreted LPS molecules, which do not assemble precoated with RsaA, likely resulting in discontinuities in the S-layer (Fig. 9). Additionally, regions of cell growth in *C. crescentus* often contain significant membrane curvature, which likely contribute to shear stress in the S-layer leading to lattice rupture, because geometrically, a hexagonal lattice cannot tesselate perfectly along closed curved surfaces. A pool of freely diffusible RsaA molecules is present between the OM and previously crystallised S-layer, secreted by the partner T1SS[41], evidenced by light microscopy[34], cryo-EM structures and cryo-ET of cells[28]. These unassembled RsaA molecules would always be available to plug any gaps in the lattice, caused by damage from environmental pressures or, as observed in our study, regions of cell growth and high membrane curvature. Once integrated into the S-layer, RsaA is non-diffuse, moving with the underlying membrane. While S-layer assembly and retention by the cell appears to be remarkably robust when challenged with a variety of

drugs, as seen in cryo-ET, S-layer localisation is selectively disrupted by MreB inhibition by A22 and polymyxin B inhibition of LPS. In the future, deeper understanding of the mechanisms driving S-layer localisation would be best examined by genetic manipulation of the RsaA secretion (RsaDEF) machinery and other cell cycle proteins. What this study indicates is that the insertion of new LPS into the OM is probably kinetically faster than the diffusion of RsaA to the O-antigen tip, leading to S-layer discontinuity in the observed regions (Fig. 9). Using unassembled RsaA molecules to plug S-layer gaps allows the cell to retain a nearly complete S-layer as it moves through the cell cycle, with a constant supply of RsaA.

The localisation pattern of S-layer biogenesis, as observed by our SpyCatcher-labelling, approach is remarkably similar to the localisation of several proteins key to cell division. For example, fluorescently labelled components of the divisome machinery in *C. crescentus*, such as DipM (LytM endopeptidase), FtsW (PG polymerase), FtsL (divisome-recruitment protein), and PBP3 (PG-crosslinking divisome protein), share marked similarities with new S-layer biogenesis[59,80–82]. Unlike S-layer biogenesis, the recruitment of these components is much better understood and relies on the highly-conserved prokaryotic tubulin homologue, FtsZ, which, along with several other key proteins, comprises the divisome complex in *C. crescentus*[78,81,83–85]. As successful cytokinesis requires the breaking and remodelling of the PG cell wall, many of these division proteins are associated with significant cell wall turnover and would therefore likely co-localise with new S-layer

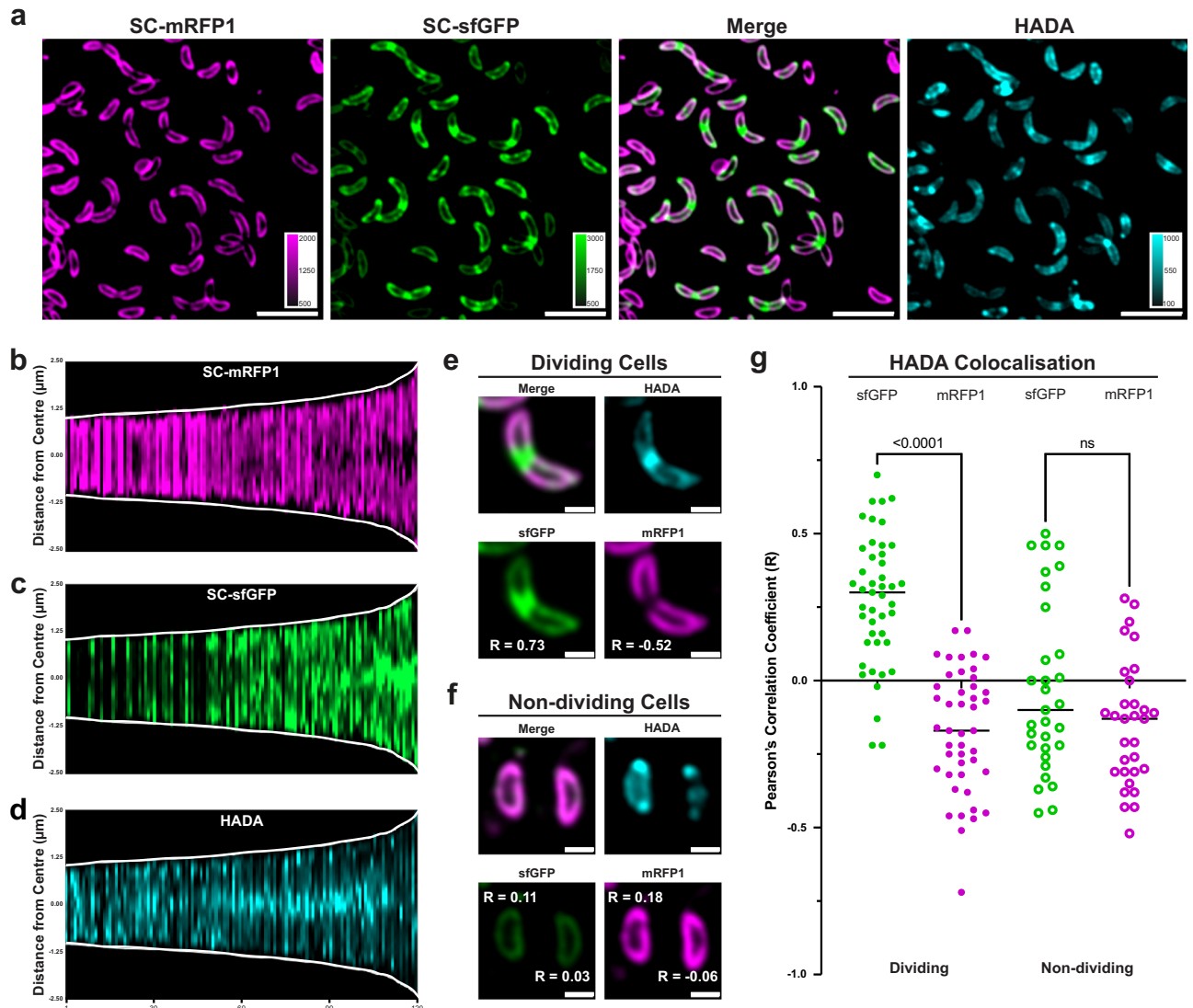

**Fig. 7 | Cell wall turnover precedes and colocalises with S-layer expansion in dividing *C. crescentus* cells. a** Three-colour labelled *C. crescentus* cells, pulse-labelled with SC-mRFP1 (magenta), chased with SC-sfGFP (green). Channels showing S-layer labelling are merged (third panel) and the HADA fluorescence shown in cyan (fourth panel). Calibration bars are provided for each channel. Scale bars = 5 μm. Demographs of labelled *C. crescentus* cells ordered by length ($n = 100$ cells), showing (**b**) SC-sfGFP, (**c**) SC-mRFP1, and (**d**) HADA fluorescence profiles for each cell. HADA signal localises to the mid-cell at shorter cell lengths than SC-sfGFP, which precedes an apparent colocalization of the sfGFP and HADA signals. The HADA signal at mid-cell eventually subsides, concurrent with further new S-layer insertion. Demograph intensities are calibrated to the same levels as in (**a**). PCC scores (R-values) for correlation of HADA with sfGFP and HADA with mRFP1 signals, comparing (**e**) dividing and (**f**) non-dividing cells. R-values are displayed next to their respective cells in the relevant channels. Scale bars = 1 μm. **g** PCC scores (R-values) between HADA and sfGFP or mRFP1 channels for dividing ($n = 45$) and non-dividing ($n = 31$) cells measuring colocalization (horizontal lines signify mean of the dataset). Dividing cells show a significantly higher R-value between HADA and sfGFP (●) compared to mRFP1 (●) (measured by two-tailed Student's *t* test, $p < 0.0001$ for dividing cells and ns for non-dividing cells), suggesting stronger colocalisation. Non-dividing cells showed a negative PCC R-value on average for HADA correlation between both sfGFP (○) and mRFP1 (○) signals, with no significant difference (measured by two-tailed Student's *t* test, $p = 0.070$). Source data are provided as an accompanying Source Data file.

insertion[60,86–88]. Whether there is a higher-level coordination of S-layer biogenesis beyond local cell envelope expansion, for example through cytoskeletal localisation, is difficult to determine. Past studies on cell division and our results here suggest at least a transient co-ordination of multiple envelope components in *C. crescentus* (Fig. 9). There is clear tendency for many bacterial and archaeal cells to synchronise the biogenesis of different envelope components[4], which appears to be the case in *C. crescentus* as well.

While it is tempting to suggest that unbound LPS at the observed regions of discontinuous S-layer is freshly secreted, remarkably little is known regarding the potential localisation of LPS integration into the OM in *C. crescentus*. In two Gram-negative bacteria *Brucella abortus*

and *Agrobacterium tumefaciens* that exhibit polar growth[89], localisation of the LPS-biosynthetic machinery to regions of cell-growth has been demonstrated[90]. Additionally, previous studies have shown that the OM is diffusion restricted and that the OM composition is directly regulated by cell wall turnover in bacteria such as *Escherichia coli*[91]. Polymerised *C. crescentus* S-layer (but not monomeric RsaA), is also diffusion-restricted (Supplementary Fig. 1), so it is possible that these SLP-deficient OMs and associated LPS may also have been recently inserted in *C. crescentus*.

Owing to the crystalline nature of the *C. crescentus* S-layer, SLPs (RsaA molecules) are likely able to self-integrate themselves into the growing lattice at gaps in the two-dimensional crystal. This has been

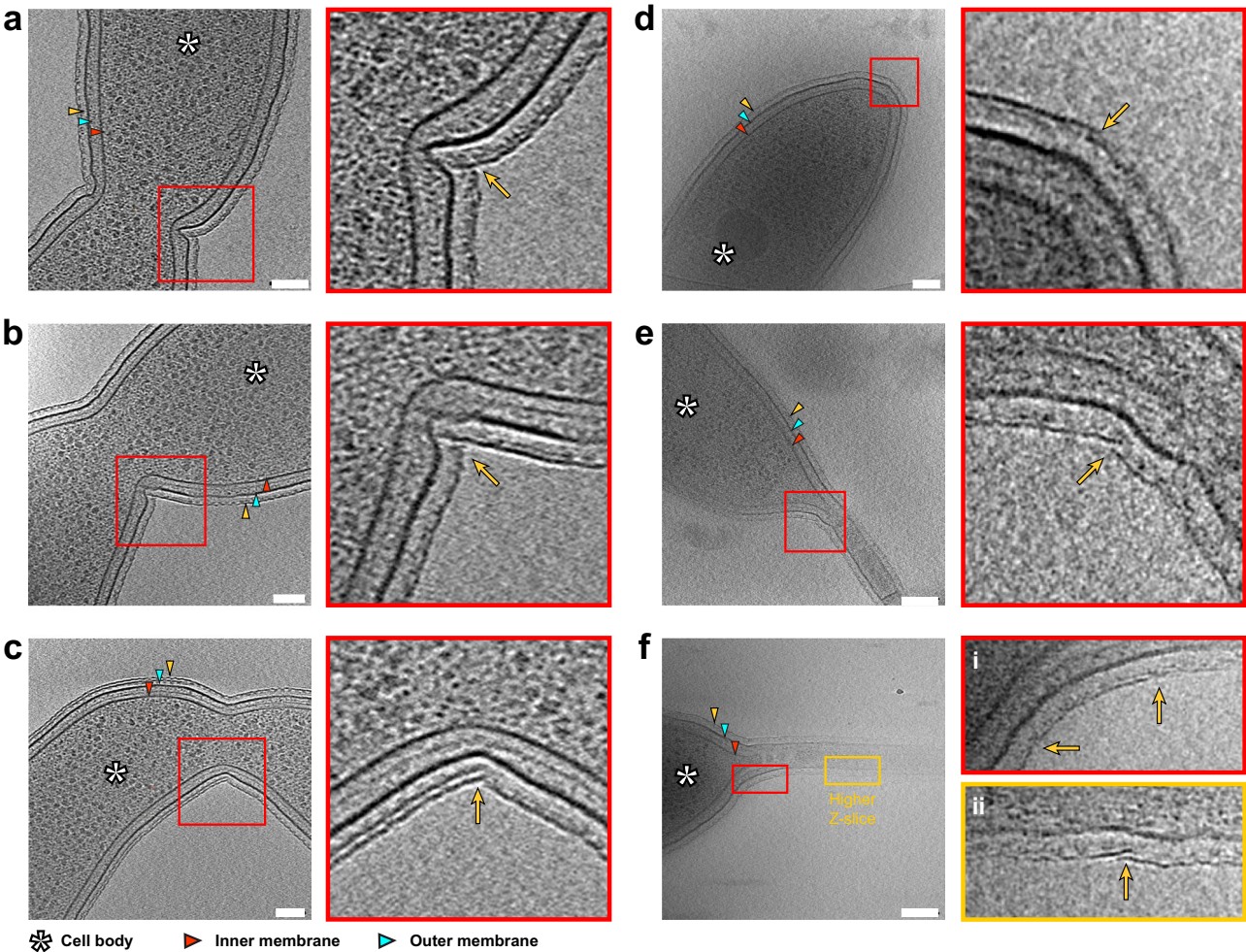

**Legend:**
- ✳ Cell body
- ▶ (red) Inner membrane
- ▶ (cyan) Outer membrane
- ▶ (yellow) S-layer
- ⟹ (yellow) Lattice break/gap

**Fig. 8 | S-layer insertion events at regions of cell growth and high membrane curvature identified in our light microscopy experiments. a–f** Slices through tomograms of *C. crescentus* cells and zoomed views (defined by the red square) showing possible insertion events. Insertion events cover regions of S-layer biogenesis, as established by our light microscopy. **a–c** Mid-cell discontinuities of the S-layer. **d** Flagellate cell pole (flagellum visible at lower Z-slices). **e**, **f** *C. crescentus* polar stalks. Components of the cell enveloped have been labelled. Specifically, yellow arrows denote regions of discontinuities or overlaps in the S-layer lattice. Cases where the Z-slice have been changed for the zoomed panel have been marked. Scale bars = 100 nm.

observed in vitro, where purified RsaA in the presence of calcium spontaneously forms hexameric lattices comparable to those seen on the cell surface[29,30,92]. This presents an ingenious solution to a difficult logistical problem of the polymerisation of the highly ordered S-layer, requiring no further energetic input from the cell beyond secretion of the constituent SLP into the extracellular milieu, where it will bind the surface and oligomerise. It is remarkable therefore, that other studied S-layers with a similar mid-cell insertion phenotype, do not possess extensive gaps in the lattice, but rather complete the S-layer with defined geometric defects[93–95]. For example, in the archaeal S-layer of *Haloferax volcanii*, pentamers and heptamers were observed on cells, which were coated to near-perfect continuity by the hexagonal S-layer[12]. Geometrically, to close a hexagonal sheet, defects or gaps must be present, therefore it will be intriguing to study why different organisms have adopted different solutions to this problem. Future research in this direction will illuminate our understanding of curved lattices in cells, which are ubiquitous across domains of life.

Our study did not localise the RsaA secretion machinery, the type 1 secretion system (T1SS) RsaDEF, components of which share homology with a variety of other T1SS machineries[38,41,42,44,96–98]. While principles of egress by T1SS have been extensively investigated, there is little literature on the localisation of T1SS across Gram-negative

bacteria[99–101]. Interestingly, in *C. difficile*, the secretion machinery of the major structural SLP SlpA, called SecA2, was localised using a modified, fluorescent version of SlpA blocked for secretion by an N-terminally linked dihydrofolate reductase[33]. This study found that while SecA2 was localised all along the cell surface, SlpA integrated into the growing S-layer at regions of cell wall turnover. These findings are consistent with our proposed model for S-layer biogenesis, despite *C. crescentus* and *C difficile* having strikingly different cell cycles and envelope organisations (diderm versus monoderm, respectively). In contrast, the S-layer assembly machinery of *H. volcanii* appears to be consistently localised with regions of new S-layer biogenesis[32], which is also colocalised with FtsZ. This difference in spatial localisation of secretion vs. biogenesis may arise from the fact that in *H. volcanii*, the S-layer glycoprotein is lipidated, attached directly to the membrane and that the S-layer constitutes the major structural component of the cell envelope[22,102]. For the *C. crescentus* S-layer, future studies on the localisation of RsaDEF using genetic mutants would provide further context into the mechanisms of S-layer secretion, and how this compares to other species.

S-layers are widespread in prokaryotes, but fundamental biology related to S-layers is poorly understood. This cell biology study of S-layer insertion in *C. crescentus* attempts to address an important gap

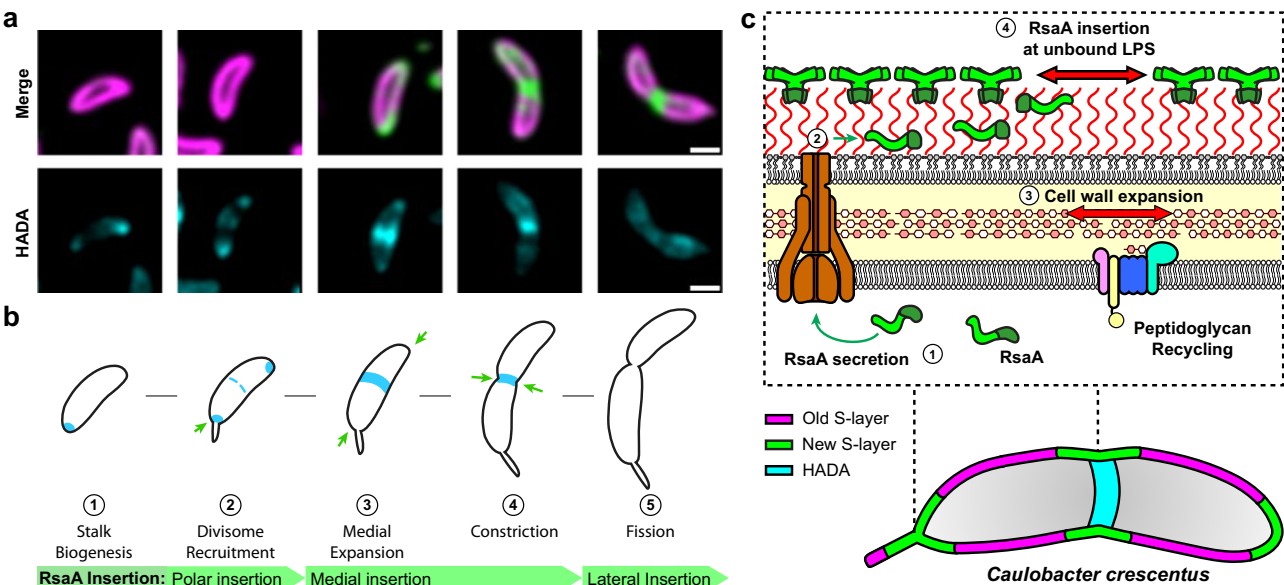

**Fig. 9 | Proposed model of cell-expansion dependent S-layer insertion in *C. crescentus*. a** Micrographs showing individual *C. crescentus* cells of varying length, beginning with a non-dividing cell and ending with pre-divisional cell. Top row shows merged SC-mRFP1 and SC-sfGFP signals, bottom row shows HADA signal. Scale bars = 1 μm. HADA labelling experiments were repeated at least four times with comparable results. The micrographs presented are representative of a single experimental repeat. **b** Schematic representation of the relationship between re-localisation of S-layer biogenesis to regions of cell wall turnover and expansion (green arrows). **c** Model of S-layer biogenesis in *C. crescentus*; unfolded, monomeric RsaA is exported from the cytoplasm into the extracellular space, whereupon it binds LPS. RsaA then diffuses along the LPS until it finds a region of surface expansion corresponding to a gap in the S-layer, driven by PG turnover and poly-merisation in the cell wall, leading to RsaA binding to the tip of the LPS and oligomerising with the pre-existing S-layer to complete the lattice.

in our knowledge and will help place future studies on S-layers into context. Our results into S-layer biogenesis will also be of great interest to microbiologists studying cell division and the cell cycle because S-layer biogenesis appears to be tightly linked to the cell cycle in many organisms across domains of life[4,32,33,95,103]. In addition, S-layers are purported to be one of the earliest and simplest attempts of cells to construct a cell wall[1]; the energy efficient mechanisms by which they oligomerise and bind their underlying cells represent an intriguing solution to the many challenges the cell surface must address[31].

Moreover, our studies have implications in the design and utilisation of S-layers as platforms for synthetic biology applications[104]. The *C. crescentus* S-layer has already been adapted as an engineering hub for purposes ranging from protein display[96] to customisable scaffolds for the design of living materials[45,105]. The exploitation of natural systems for the development of engineered living materials has broad applications[106]. In particular, the self-organising and self-healing nature of S-layers provides an ideal scaffold for the design of a high-density biological interaction[45,105]. We expect that uncovering the mechanisms of spatiotemporal localisation of S-layer biogenesis will help further harness the *C. crescentus*' S-layer as a platform for synthetic biology. Particularly, our demonstration of the consistent and predictable nature of RsaA integration into the expanding S-layer, supports fine tuning of protein scaffolding on the cell. These implications on synthetic biology, together with the improved insight into fundamental microbiology highlight why vital future research is needed to understand these captivating S-layer arrays found in abundance in prokaryotes.

## Methods

### SpyCatcher purification
His-tagged SpyCatcher conjugates were purified as previously described using nickel-affinity chromatography[31]. Briefly, plasmids pDEST14-SpyCatcher-sfGFP and pBAD-SpyCatcher-mRFP1 were transformed into chemically competent *E. coli* BL21 (DE3) and LMG194 cells respectively and grown on LB agar supplemented with 100 μg/mL

Ampicillin (LB-Amp). A single colony of each strain was inoculated into 6 L of LB-Amp media and incubated at 37 °C with shaking until cells had reached mid-log growth phase. Cells were induced with 0.2% (w/v) arabinose (LMG194) or 0.4 mM Isopropyl β-D-1-thiogalactopyranoside (IPTG) (BL21) and incubated with shaking at 20 °C for 16 h. Induced cultures were harvested by centrifugation (15,000 relative centrifugal force (x g), 30 min, 4 °C), resuspended in lysis buffer (30 mM Tris/HCl pH 8.0, 500 mM NaCl, 1 mM MgCl₂, 50 μg/mL DNase, 300 μg/mL lysozyme, and 1x cOmplete Protease Inhibitor), and lysed by five passes through the homogeniser at 15,000 psi (pounds per square inch) pressure. Cell debris were pelleted (50,000 (x g), 45 min, 4 °C), and the supernatant filtered using a 0.22 μm syringe filter. SpyCatcher proteins were then bound to a 5 mL HisTrap HP column (GE Healthcare) using an ÄKTA pure 25 M system (GE Healthcare) and eluted against the same buffer including 500 mM imidazole over 10 column volumes. Eluates were dialysed overnight with 1:100 (w/w) His₆-TEV protease at 4 °C against 2 L of MilliQ H₂O. The dialysates were further purified via size exclusion chromatography using a HiLoad Superdex S200 16/600 (prep grade) column; final proteins were eluted in HEPES buffer (25 mM HEPES/NaOH pH 7.5, 150 mM NaCl), and flash frozen in liquid nitrogen and stored at −80 °C.

### SpyCatcher and HADA labelling of *C. crescentus*
*C. crescentus* expressing RsaA-467-SpyTag (CB15N Δ*sapA rsaA467*:S-pyTag) cells were grown in PYE media (0.2% (w/v) Bacto Peptone, 0.1% (w/v) yeast extract, 0.5 mM CaCl₂, 1 mM MgSO₄) at 30 °C with aeration by shaking to mid-log growth phase. For SpyCatcher labelling, cells were resuspended to OD₆₀₀ 0.1 in PYE, followed by pulse labelling with 10 μM SC-mRFP1 at 4 °C for 16 h, after which point cells were harvested by centrifugation (3 min, 8000 × g) and washed three times with chilled PYE. For chase labelling, cells were resuspended in fresh PYE media and incubated at 30 °C for 1.5 h in the presence of 10 μM SC-sfGFP to stimulate growth. After labelling, cells were harvested by centrifugation and washed as described above, followed by resuspension in a final volume of 50 μL PYE. For PG labelling, cells were supplemented

with 500 μM of the fluorescent D-amino acid HADA[107] (Cambridge Biosciences) for the last 10 min of the chase-labelling incubation. Cells were harvested and washed (as above). When required, cells were resuspended in chilled 4% formaldehyde (in PBS) for fixation. Samples were kept at 4 °C for 20 min prior to washing (as above) and imaging. All incubation steps were carried out with the specimen protected from light-exposure.

When necessary, cells were synchronised using density centrifugation method using colloidal silica as follows[108]. Cells were grown and pulse labelled by incubation overnight with SC-mRFP1, as described. Cells were washed three times with PBS and resuspended in 750 μL ice-cold PBS. Samples were mixed 1:1 with syringe-filtered 33% chilled Percoll (Sigma-Aldrich), then centrifuged at 15,000 × g in a tabletop centrifuge for 20 min at 4 °C, separating the cells into top (stalked cell) and bottom (swarmer cell) bands. The top band was carefully removed, and the bottom band collected, in a final volume of 50–200 μL depending on the band size. Swarmer cells were pelleted and washed three times in ice-cold PBS media to remove excess Percoll. Cells were then resuspended in fresh PYE and chase labelled using SC-sfGFP as described above.

### SpyCatcher labelling of the drug treated *C. crescentus*
To select sub-lethal concentrations of the selected drugs for subsequent pulse-chase labelling experiments, growth curves of *C. crescentus* were generated at various concentrations of the drugs by measuring optical density (OD at 600 nm wavelength of light) of cultures with hourly intervals. Following analysis of the growth curves, the selected sub-lethal concentrations were as follows: 50 μg/mL for cephalexin, 7.5 μg/mL for polymyxin B, 0.5 μg/mL for chloramphenicol, 3 μg/mL for A22 and 0.5 μg/mL for MMC. For the drug treated cells, pulse-chase labelling experiments were performed as described above except for the SC-sfGFP chase incubation step, which was extended to 3 h.

### Light microscopy
Two μL of labelled cell suspensions were spotted onto agarose pads (1% (w/v) in distilled water) enclosed by a 15 mm × 16 mm Gene Frame (ThermoFisher) on a glass slide and sealed with a glass coverslip. For cells labelled using only SpyCatcher conjugates, cells were imaged using an Olympus SoRa spinning disc confocal microscope, equipped with Olympus IX-83 inverted frame, Yokogawa SoRa super-resolution spinning disc module, and Prime BSI camera. Slides were kept at room temperature and imaged using the 60x (1.5 NA) lens, with excitation at 488 nm (sfGFP) and 561 nm (mRFP1) (solid state lasers), 200 ms exposure. Z-stacks were taken at 0.26 μm intervals and an Olympus Super Resolution filter was applied to the entire stack using Olympus CellSens software, followed by deconvolution using a maximum likelihood algorithm (5 iterations). In general, Z-stacks were condensed using a maximum Z-projection of frames containing the cell of interest (±1 frame on the upper and lower Z-axis). For HADA-labelled samples, cells were imaged using an Olympus Fluoview FV1200 equipped with equipped with GaAsP detectors. Images were acquired using a 100 x (1.4 NA) lens with excitation via solid state 405 nm (HADA), 559 nm (mRFP1), and argon 488 nm (sfGFP) lasers, with scanning at 1024 × 1024 pixels. A Kalman filter (2 iterations) was applied for all image collections. Single slices through the middle of the cells were taken, without Z-stacks, to limit photobleaching of the sample. Images were background-subtracted and filtered using a 0.5-pixel Gaussian blur (ImageJ) unless stated otherwise. For the drug treated samples, cells were imaged using a Nikon W1 spinning disc confocal microscope with spinning disc module, and sCMOS camera (95% QE). For the drug-treated samples, slides were kept at room temperature and imaged using the 100x (1.4 NA) lens, with excitation at 488 nm (sfGFP) and 561 nm (mRFP1).

### Light microscopy image analysis
Demographs and cell intensity profiles were generated using the MicrobeJ 5.13 plugin for ImageJ/FIJI[46]. Cell debris, overlapping cells, or cells on the edge of the micrographs were excluded from the analysis. The remaining cells were normalised for fluorescence intensity and plotted according to length from shortest to longest. Cell intensity profiles are representative of 100 dividing and non-dividing cells, assigned by the presence of invagination at the mid-cell, from the demograph. Normalised profile intensities from the sfGFP and mRFP1 channels, including standard deviation, were plotted relative to the cell length. For individual cell-profile analyses, a line was manually drawn along the indicated region of the cell through the cell body, straightened, and the pixel values extracted. Intensity values were normalised for each channel and plotted relative to the cell contour. Where given, cell profiles were binarized according to the presence of the strongest fluorescence intensity value. For colocalisation studies, masks were created for individual cells in ImageJ and colocalisation measured using PCC in the Coloc2 plugin. A mask was created for individual cells and the colocalisation of the RFP1- and sfGFP-labelled regions was measured by PCC. All micrograph images were generated in ImageJ/FIJI.

### Electron cryotomography (cryo-ET) sample preparation, data collection and analysis
Cryo-ET grid preparation was performed as described previously[31,78,109]. Briefly, 2.5 μL of the relevant *C. crescentus* cell sample (OD$_{600}$ 0.5–0.7 in PYE or M2G) mixed with 10 nm protein-A gold (CMC Utrecht) was applied to a freshly glow discharged Quantifoil R2/2 or R3.5/1Cu/Rh 200 mesh grid, adsorbed for 10 s, blotted for 2.5 s and plunge-frozen into liquid ethane in a Vitrobot Mark IV (ThermoFisher), while the blotting chamber was maintained at 100% humidity at 10 °C. For tomographic data collection, the SerialEM 3 software[110] was used as described previously[111], using the Quantum energy filter (slit width 20 eV) and the K2 or K3 direct electron detector running in counting mode. Tilt series with a defocus range of −5 to −8 μm were collected between ±65° in a bidirectional (only for Fig. 8a–c) or ±60° dose symmetric scheme with a 1° tilt increment. A total dose of 150 e$^-$/A$^{°2}$ (only Fig. 8a–c) or 73 e$^-$/A$^{°2}$ was applied over the entire series. Cryo-ET data analysis was performed in IMOD[112] and tomographic reconstruction was carried out using the SIRT algorithm implemented within Tomo3D[113,114]. Datasets were motion corrected and dose weighted with MotionCor2[115] implemented in Relion 3.0[116]. Contrast transfer functions (CTFs) of the resulting motion corrected micrographs were estimated using CTFFIND4[117]. Tomogram slices for visualisation were prepared using USCF Chimera[118] or ChimeraX 1.4[119].

### Additional quantification and statistical analyses
Graphs, with the exception of cell profile plots and demographs, were generated in and statistical analyses were carried out using GraphPad Prism 9.5.1. Where possible, the number of measurements performed are reported within the relevant figure legends.

### Reporting summary
Further information on research design is available in the Nature Portfolio Reporting Summary linked to this article.

## Data availability
The source data underlying Figures with plotted graphs is available with this manuscript as a Source Data file. All other data, including cryo-ET and light microscopy images, are available from the authors upon request. Source data are provided with this paper.

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

## Acknowledgements

M.H. was supported by funding from the Biotechnology and Biological Sciences Research Council (BBSRC, grant number BB/M011224/1). T.A.M.B. is supported by the Medical Research Council, as part of United Kingdom Research and Innovation (also known as UK Research and Innovation) [Programme MC_UP_1201/31]. For the purpose of open access, the MRC Laboratory of Molecular Biology has applied a CC BY public copyright licence to any Author Accepted Manuscript version arising. The authors would like to thank the MRC LMB light microscopy facility. T.A.M.B. would like to thank the Wellcome Trust (Discovery Award 225317/Z/22/Z), the Human Frontier Science Program (Grant RGY0074/2021), the Vallee Research Foundation, the European Molecular Biology Organization, the Leverhulme Trust, and the Lister Institute for Preventative Medicine for support. We thank Jan Löwe and Buzz Baum for critically reading this manuscript.

## Author contributions
Experiments were conceived and designed by M.H. and T.A.M.B. Technical expertise and support for the collection and analysis of light microscopy data was provided by U.S. and A.W., M.H., B.I., A.V.K. and T.A.M.B. were responsible for data collection and analysis. M.H. wrote the manuscript and created the accompanying figures, and all authors contributed equally to reviewing and editing the manuscript.

## Competing interests
The authors declare no competing interest.

## Additional information

Tanmay A. M. Bharat.

