## [Peer Review File · Nature Communications]

Cell Cycle Dependent Coordination of Surface Layer Biogenesis in *Caulobacter crescentus*Reviewer #1 (Remarks to the Author):

Little is known about S-layer expansion during bacterial growth. The authors aim to explore this problem and report the very interesting observation from SpyCatcher pulse chase fluorescence labelling experiment which is interpreted to show that new RsaA molecules are inserted at the division plane and poles. This claim would indeed be exciting, but after this single experiment the study is not developed substantially further and the experimentation/data is not clear and ambiguous. My recommendation is to bolster the observations with deeper mechanistic insight.

First, while the pole and division plane localization certainly suggests that there is new insertion of RsaA molecules at these sites, in principle it is also possible that RsaA-spy is only accessible to SpyCatcher at these sites and that access to SpyCatcher is cell cycle regulated, not the insertion or the recruitment of RsaA-Spy. SpyCatcher is a fairly large fluorescent molecule that could be obstructed from access (in theory) to RsaA by outermost envelope layers such as the capsule. *Caulobacter* does have a capsule and it is not correct to state (on line 21 in the abstract) that the S-layers are the outermost envelope layers. Is SpyCatcher fluorescence different when cells are fixed and permeabilized before the labelling?

A chase experiment in which protein synthesis is arrested with chloramphenicol before the second SpyCatcher label is added might also clarify this question, and eliminate the possibility of diffusion to this site because it could represent a region of the cell where RsaA molecules are highly fluid or the S-layer is not fully compact. This would help differentiate between the site of RsaA secretion versus retention at the poles/midcell. This question also brings up the major point that goes hand in hand with the model presented here: localisation of the RsaA secretion machinery and I think this needs to be reported here, especially if the ambiguous inhibitor studies with cephalixin are not bolstered further genetically (see below). If RsaA is inserted at the poles and division plane then the secretion machine should be localized at this site as well. It would make sense that the RsaA secretion machine, an envelope spanning system that must penetrate the PG, would assemble at sites where plenty of PG remodeling occurs since, PG penetration of this large protein machine would be easier at this site.

Is de novo RsaA localisation at the poles and division sites dependent on septal PG and/or the divisome? This part of the data is very ambiguous because the cephalixin inhibition did not work well. *Caulobacter* has a beta-lactamase that cleaves cephalixin so high amounts are needed and the inhibition is only partial: after treatment cells are bit longer but have poles and septa. The SpyCatcher pulse labeling experiments are not clear and the results should be bolstered by getting smooth filaments by FtsZ inhibition, directly or indirectly. There are many ways in which this can be achieved in *Caulobacter*, not only by FtsZ depletion strains, but also by transient expression of FtsZ inhibitors. This generates smooth filaments, not constricted ones as seen with cephalixin that still make septal PG, and it would be easier and cleaner to conduct the SpyCatcher pulse chase experiments in this setting. The bottom line is that no convincing conclusions are possible at this stage about Is SpyCatcher labelling and the role of septal PG and/or the divisome in this RsaA-Spy pattern.

I don't know what the A22 experiments really add. Sure, A22 inhibits MreB and the cells become lemon-shaped because of PG disorganization after MreB is inhibited, but in terms of the model except perhaps adding that PG perturbations in general can influence the polar/midcell disposition of RsaA -Spy, I don't see any value. Except that the A22 inhibition worked, which cannot be said for cephalixin which the authors used in hopes of inhibiting PBP3 (FtsI).

What is the difference between fig 2B and 3A and should fig4 not be embedded in Fig 1?

are the gaps in the RsaA lattice enlarged when cell division is inhibited in FtsZ depletion strain for example? I don't see the connection between the RsaA lattice gaps and the pole/midcell

SpyCatcher labelling experiments. It seems disjointed as two independent observations without causality in this write-up.

Finally it would be very interesting to investigate if SpyCatcher labelling is affected when inhibit LPS synthesis is perturbed with CHIR-090 (LpxC inhibitor) or polymyxin which targets lipidA directly. These inhibitor experiments might work better than the cephalexin (or other beta-lactam) inhibition studies, unless it is done in mutant background that lacks the beta-lactamase or after prior chemical inhibition of the beta-lactamase.

The discussion mentions that these findings have implications on synthetic biology approaches using the S-layer as nanodevice, but is no clear to me how the RsaA pulse-chase localization which still ultimately gives a homogenous distribution of RsaA on the surface, could influence use of the S-layer. Please explain this point!

Reviewer #2 (Remarks to the Author):

Herdman et al. report on the localisation of S-layer assembly in the model bacterium *Caulobacter crescentus* and its coordination with cell envelope expansion. To do so, the authors employ pulse-chase fluorescence labelling of a Spy-tagged RsaA derivative, and follow de novo S-layer synthesis using light microscopy and cryoET. This is done both under physiological conditions as well as in conditions where cell division or MreB-coordinated cell elongation are specifically inhibited.

The study finds that S-layer assembly occurs at sites of cell envelope expansion, where it follows on de novo peptidoglycan synthesis.

The observation of localised S-layer expansion at sites of cell growth and division is in itself not new. Comerchi et al. *Nature Communications* volume 10, 2731 (2019), used a very similar pulse-chase fluorescence labelling protocol, as well as the A22 and Cephalexin-inhibited cell elongation or division, respectively. This current study differs in also labelling de novo peptidoglycan synthesis (i.e. using HADA), and in obtaining a molecular resolution view of the S-layer expansion sites by cryoET. The latter illustrates the presence of defect or dislocation zones from which the S-layer is likely to expand.

The work is well executed and results are beautifully documented. Whilst the cryoET imaging and the HADA labelling do provide a very detailed illustration of the S-layer expansion sites, the fundamental advance of this study over Comerchi et al. 2019 remains limited.

The authors note that the site of RsaA secretion remains unknown. It would be interesting to try resolve this by expression of a fluorescent RsaA fusion protein, and to see if secretion is in any way coordinated with cell expansion. I.e. would the use of A22 result in the build-up of RsaA monomers trapped underneath the S-layer? Is this what the authors are illustrating in Fig. 3E, i.e. the build-up and eventually crystallisation of RsaA underneath the S-layer? The text and legend are not very clear on this.

Furthermore, now with the structure in hand, did the authors consider using an assembly incompetent mutant of RsaA, or just the cell-anchoring domain to see if these would compete with the sites of S-layer expansion. I.e. is it the availability of free LPS sites or the availability of a sterically free crystals growth edge that become limiting to expansion of the S-layer?

Minor points.

In line 210 and later that paragraph lines 218-19 the authors describe the presence of 'gaps' in the S-layer. The images appear to show defects or dislocation of the S-layer growth edge more than

showing gaps, so that this terminology may be somewhat misleading.

Reviewer #3 (Remarks to the Author):

This manuscript by Herdman et al describes a series of light microscopy and cryo-tomography experiments focused on the assembly of new S-layer on the surface of *Caulobacter crescentus*. The experiments are very well-designed, the quality of the data is high and the interpretation is reasonable and supported by the presented data. Overall there's very little to criticise here and I commend the authors on an elegantly designed study and well-written manuscript.

The work presented here is largely in agreement with previous studies of S-layer growth in multiple species. These are cited in the introduction, but the parallels are largely ignored in the discussion unfortunately. It would be useful to more fully place these new *Caulobacter* findings in the context of what is already known. I think this is particularly important in light of the observation of boundary defects and gaps at the site of insertion. The authors point out that this has not been observed previously but might this reflect the resolution of the methods used in previous studies? Is it possible that this is a more common feature of bacterial S-layer growth?

Also, given the authors previous description of a pool of free subunits underlying the S-layer can you speculate about what these defects tell us about the kinetics of assembly? Why are these gaps not efficiently filled as the underlying cell envelope grows?

In other species distinct grain boundaries have been observed in the S-layer and it is thought that these form as a consequence of S-layer growth. Are there symmetry mismatches observed across the gaps observed here? Is lattice orientation consistent across the *Caulobacter* surface? If so that would seem to be an important point to highlight.

Minor issues:

Line 43: abundant is an odd choice of word here

Line 98: different terminology used to describe the cell types and in the figure here (swarmer vs non-dividing) – would be helpful to the non-specialist reader to define these more clearly and use a single nomenclature.

Line 228: typo "at the several areas"

Line 259: typo "the biogenesis different"

Response to reviewers' comments for Herdman *et al*

Black text – reviewer comment

Red text – author response

Blue text – manuscript excerpt

We thank the reviewers for their advice. In light of the reviewers' constructive comments and suggestions, we have carried out a number of additional experiments to strengthen the presented research. This constitutes a major revision of the manuscript.

Reviewer #1 (Remarks to the Author):

Little is known about S-layer expansion during bacterial growth. The authors aim to explore this problem and report the very interesting observation from SpyCatcher pulse chase fluorescence labelling experiment which is interpreted to show that new RsaA molecules are inserted at the division plane and poles. This claim would indeed be exciting, but after this single experiment the study is not developed substantially further and the experimentation/data is not clear and ambiguous. My recommendation is to bolster the observations with deeper mechanistic insight.

We thank the reviewer for their encouraging comments. We have bolstered the manuscript with extensive further experiments, analyses and text as suggested.

First, while the pole and division plane localization certainly suggests that there is new insertion of RsaA molecules at these sites, in principle it is also possible that RsaA-spy is only accessible to SpyCatcher at these sites and that access to SpyCatcher is cell cycle regulated, not the insertion or the recruitment of RsaA-Spy. SpyCatcher is a fairly large fluorescent molecule that could be obstructed from access (in theory) to RsaA by outermost envelope layers such as the capsule. *Caulobacter* does have a capsule and it is not correct to state (on line 21 in the abstract) that the S-layers are the outermost envelope layers. Is SpyCatcher fluorescence different when cells are fixed and permeabilized before the labelling?

The reviewer is correct in that our labelling methodology is only capable of binding accessible RsaA-SpyTag sites. To prevent any misunderstanding, we have added some text to highlight this in the main text where the labelling is introduced -

Line 98: "Following labelling of the surface available SpyTags, we observed *C. crescentus* cells with distinct fluorescent regions of mRFP1- and sfGFP-labelling,..."

However, if it were the case that SpyCatcher proteins preferentially bound the mid-cell and cell poles due to space restriction, we would expect this to be evident following the initial pulse labelling; instead, we observed uniform labelling across the cell surface that is only disrupted by allowing the cells to grow. This is also apparent when sub-saturating concentrations of SpyCatcher conjugates are used; uniform labelling across the cell surface with no bias to the mid-cell or cell poles (see below, not included in the manuscript).

Effect of Saturating vs. Sub-Saturating Concentrations of SC-mRFP1.

C. crescentus expressing RsaA-467-SpyTag was incubated with 10 μM (top row) and 1 μM SC-mRFP1, 4 $^{\circ}\text{C}$ with shaking for six hours. Brightfield and SC-mRFP1 channels were used to image the cells using the same conditions and are presented in the same contrast. The micrograph of cells labelled in the presence of 1 μM SC-mRFP1 has been duplicated and presented in enhanced contrast to make labelling more visible, showing consistent labelling in the entire cell surface.

With regards to our inaccurate assertion that the S-layer is the outermost component of the cell envelope, we agree with the reviewer’s point that many prokaryotes possess an extracellular capsule that is part of the envelope. The text has been updated to acknowledge this -

Line 19: “Surface layers (S-layers) are a family of proteinaceous, two-dimensional crystalline arrays that comprise a major component of many prokaryotic cell envelopes.”

and

Line 287: “Given their position as one of the outermost components of the envelope, coating the entire cell surface”

We have not tested the levels of fluorescence in permeabilised and non-permeabilised cells. However, the duration of the initial pulse labelling and the concentration of SpyCatcher used (100:1 molar ratio of SpyCatcher to RsaA-SpyTag) is far in excess of the conditions required to achieve saturation of the available SpyTag peptides, as described in (Charrier et al., 2019). Additionally, the capsule of *C. crescentus* is enriched in swarmer cells, and we observe no difference in fluorescence intensity or profile (except, of course, in the localisation pattern of old and new S-layer) between swarmer and stalked cell types.

A chase experiment in which protein synthesis is arrested with chloramphenicol before the second SpyCatcher label is added might also clarify this question, and eliminate the possibility of diffusion to this site because it could represent a region of the cell where RsaA molecules are highly fluid or the S-layer is not fully compact. This would help differentiate between the site of RsaA secretion versus retention at the poles/midcell.

We have performed this experiment as suggested (new Supplementary Fig. 3, below). Chloramphenicol inhibits protein synthesis; we found it reduced cell growth, but there was no observable impact on the localisation of new S-layer insertion. The fluorescence intensity was a bit weaker, but overall similar to that of untreated *C. crescentus* cells, showing separate patches of old and new S-layer insertion (see figure below). This suggests that the appearance of the gaps is independent of RsaA translation in the cytosol.

“Supplementary Fig. 1. *C. crescentus* growth curves under exposure to various compounds. *C. crescentus* *rsaA-467-spytag ΔsapA* grown in the presence of varying concentrations of (a) mitomycin C (MMC), (b) polymyxin B, (c) cephalixin, (d) A22, and (e) chloramphenicol at various concentrations to determine sub-lethal concentrations for pulse-chase labelling experiments. All samples were grown in liquid PYE media at 30 °C, with shaking. (f) *C. crescentus* cells grown in varying concentrations of chloramphenicol over 5 hours. Pulse-chase labelled in SC-mRFP1 (magenta) and SC-sfGFP (green) in PYE (no treatment) or PYE supplemented with 0.5 µg/mL chloramphenicol. Scale bars = 5 µm.”

and

Line 225: “Moreover, inhibition of protein synthesis by the addition of chloramphenicol (Supplementary Fig. 3) also did not disrupt S-layer localisation, suggesting that targeting of old and new S-layer to their respective sites in the cell is not dependent on translation.”

To further address the concerns that the labelled regions are a result of fluidity or compaction of the S-layer, we conducted live-cell imaging experiments using SC-mRFP1-labelled *C. crescentus* cells. The non-fluorescent regions remain non-diffuse and bound to same area of the cell surface (new Supplementary Fig. 2).

“Supplementary Fig. 2. Live cell imaging of SC-mRFP1 labelled *C. crescentus* cells.

(a-c) *C. crescentus* cells labelled using SC-mRFP1, grown on PYE-agar at 30 °C. Cells were imaged over 1 hour in 30 second intervals. Cell growth and progression of the cell cycle was followed using the kymographs provided for each cell, showing fluorescence along the medial axis. Fluorescent regions remain uniform in size throughout the labelling, consistent with the proposed non-diffusible nature of crystallised RsaA on the cell surface.”

We would also like to point out that the S-layer of *C. crescentus* has been previously shown to be diffusion-restricted after incorporation into the growing S-layer (Comerci et al., 2019). In addition, we observe a confluent RsaA lattice in all our tomograms, with the exception of the small faults or defects presented in the manuscript. A more diffuse or less compacted S-

layer would be immediately apparent due to a loss of the striking organisation of the RsaA ultrastructure. We have updated the text to highlight this -

Line 117: “The LPS-bound S-layer has been previously shown to be non-diffusing relative to the underlying cell following oligomerisation(Comerci et al., 2019). To confirm this, we labelled the cell surface with SC-mRFP1, and imaged the cells live, during growth at 30 °C. As expected, we observed the same localisation of the old, SC-mRFP1-labelled S-layer on the cell body, away from the mid cell and cell poles (Supplementary Fig. 2). As the cells grew, these regions remained stationary and non-diffusing relative to the cell, evident in kymographs of growing cells.”

This question also brings up the major point that goes hand in hand with the model presented here: localisation of the RsaA secretion machinery and I think this needs to be reported here, especially if the ambiguous inhibitor studies with cephalexin are not bolstered further genetically (see below). If RsaA is inserted at the poles and division plane, then the secretion machine should be localized at this site as well. It would make sense that the RsaA secretion machine, an envelope spanning system that must penetrate the PG, would assemble at sites where plenty of PG remodeling occurs since, PG penetration of this large protein machine would be easier at this site.

We agree with the reviewer that it is most logical for secretion of RsaA and its insertion into the growing S-layer to be co-localised. We have attempted to localise the RsaDEF machinery using a genetic tagging approach in collaboration with Yves Brun and Maxime Jacq (see below). Unfortunately, the genetic tagging led to deficiency in RsaA secretion. It is known that any modifications to this system can spontaneously lead to secretion or assembly deficiency (Nomellini et al., 2007, Bingle et al., 1997, Duval et al., 2011).

Localisation of mNeonGreen-RsaD (mNG-RsaD) in *C. crescentus*.

(A) Micrographs of *C. crescentus* cells, pulse labelled in SpyCatcher003 (non-fluorescent) and chased using SC-mRFP1. mNG-RsaD shows a heterogenous distribution of green punctae across the cell. Scale bar = 10 μ m. (B) Some cells show co-localisation of SC-mRFP1 (representing new S-layer) and mNG-RsaD at cell poles. Scale bar = 1 μ m. However, most punctae along the cell body have no associated SC-mRFP1 fluorescence signal. (C) Cryo-ET reveals *C. crescentus* cells expressing mNG-RsaD have a patchy S-layer, absent over much of the cell surface. Scale bar = 500 nm.

Even though we have tried very hard to answer this question in response to the reviewer's critique, we cannot give a clear-cut response at this stage. We hope to answer this in the next few years as genetic manipulation of the RsaDEF complex is not so straight-forward.

We would like to note that these results, while incomplete, are consistent with previous attempts to label the RsaADEF system, which reported patchy localisation for both RsaD and RsaFb (Werner et al., 2009). Under the advice of reviewer 3, we have also expanded the text on the localisation of S-layer secretion machineries of other species in the discussion and highlighted the need for further research into this aspect of S-layer biology.

Line 355: "Interestingly, in *C. difficile*, the secretion machinery of the major structural S-layer protein SlpA, called SecA2, was localised using a modified, fluorescent version of SlpA blocked by secretion by an N-terminally linked dihydrofolate reductase (Oatley et al., 2020). This study found that while SecA2 was localised across the cell surface, SlpA integrated into the growing S-layer at regions of cell wall turnover. These findings are consistent with our proposed model for S-layer biogenesis, despite *C. crescentus* and *C. difficile* having strikingly different cell cycles and envelope organisations (diderm versus monoderm, respectively). In contrast, the S-layer assembly machinery of *H. volcanii* appears to be consistently colocalised with regions of new S-layer biogenesis (Abdul-Halim et al., 2020), also colocalised with FtsZ. This difference in spatial localisation of secretion with biogenesis may arise from the fact that in *H. volcanii*, the S-layer glycoprotein is lipidated, forming an integral part of the membrane. For the *C. crescentus* S-layer, future studies on the localisation of RsaDEF would provide further context into the mechanisms of S-layer secretion, and how this compares to other species."

Is de novo RsaA localisation at the poles and division sites dependent on septal PG and/or the divisome? This part of the data is very ambiguous because the cephalixin inhibition did not work well.

We have performed the experiment with increased cephalixin, please see the answer to the next points. We agree that this aspect of S-layer biogenesis, whether or not cell wall turnover is mechanistically linked to the divisome or resultant of cell wall expansion driven by PG turnover warrants further research.

Our findings suggest that RsaA localisation is not dependent on localisation of the divisome, as cephalixin and mitomycin-c (new Fig. 5) both disrupt the progress of cell division without disrupting the localised S-layer biogenesis. We propose that the driving force behind S-layer localisation is the expansion of the cell envelope, as caused by PG turnover, explaining why we observe new S-layer being regularly inserted at the mid-cell and cell poles.

Line 163: “Despite cephalixin’s marked impact on cell morphology, cephalixin-treated cells retained a dual-labelled S-layer pattern with distinct regions of old and new S-layer (Fig. 2b and Supplementary Fig. 5). Repeating the co-localisation analysis in these cells confirmed that the old and new S-layer regions were strongly anticorrelated, almost to the same extent as untreated cells (Fig. 2d, average PCC = -0.45, no significant difference in Student’s *t*-test to untreated). The localisation pattern is also consistent between untreated and cephalixin-treated cells, with new S-layer localising to the mid-cell and cell poles (Supplementary Fig. 5a-d). The cephalixin-treated cells often showed some additional new S-layer in the cell body, equidistant between the cell poles and midcell, at potential sites of disrupted cell division.”

and

Line 213: “First, to confirm the effect of disrupted cell division on S-layer biogenesis, presented above for cephalixin (Figs. 2-3), we treated the cells with sub-lethal concentrations (Supplementary Fig. 3) of mitomycin C (MMC), a DNA-crosslinking agent (Iyer and Szybalski, 1963). MMC induces cell filamentation via the SOS response (Janion, 2008), whereby cells inhibit their own division to prevent the propagation of mutations to progeny cells. This has been well characterised in *C. crescentus* (Modell et al., 2011, Modell et al., 2014) and other bacterial species, including *Streptomyces venezuelae* (Falguera et al., 2022, Stratton et al., 2022). We also included a sample containing an increased concentration of cephalixin (100 µg/mL) to ensure the observed retention of S-layer localisation is not concentration-dependent. Both the increased concentration of cephalixin and MMC resulted in cell elongation (Fig. 5a). In addition, both conditions produced cells with an anticorrelated localisation of old and new S-layer, evidenced by strong, negative PCC R-values, comparable to untreated cells (Fig. 5b). This confirmed that inhibiting cell division does not lead to delocalisation of S-layer biogenesis.”

Caulobacter has a beta-lactamase that cleaves cephalixin so high amounts are needed and the inhibition is only partial: after treatment cells are bit longer but have poles and septa. The SpyCatcher pulse labeling experiments are not clear and the results should be bolstered by getting smooth filaments by FtsZ inhibition, directly or indirectly. There are many ways in which this can be achieved in Caulobacter, not only by FtsZ depletion strains, but also by transient expression of FtsZ inhibitors. This generates smooth filaments, not constricted ones as seen with cephalixin that still make septal PG, and it would be easier and cleaner to conduct the SpyCatcher pulse chase experiments in this setting. The bottom line is that no convincing conclusions are possible at this stage about Is SpyCatcher labelling and the role of septal PG and/or the divisome in this RsaA-Spy pattern.

We have performed the experiment with increased cephalixin as requested (new Fig. 5 and below).

“Fig. 1. Quantification of spatial separation of old and new S-layer following exposure to various compounds.

(a) Pulse-chase labelled *C. crescentus* cells following growth in PYE media with no drug (untreated), or PYE supplemented with one of four compounds: 0.5 $\mu\text{g}/\text{mL}$ mitomycin C; 100 $\mu\text{g}/\text{mL}$; cephalalexin; 7.5 $\mu\text{g}/\text{mL}$ polymyxin B. Scale bar = 5 μm . (b) Co-localisation of mRFP1 and sfGFP fluorescence was quantified using Pearson’s Correlation Coefficient (PCC). Untreated cells and those treated with cell division-inhibiting compounds mitomycin C and cephalalexin showed negative PCC scores (average R-values -0.58, -0.70, -0.45, respectively). Following exposure to the LPS lipid A binding peptide polymyxin B, cells showed positive PCC scores (average R-value 0.84), revealing a disruption of spatially localised S-layer biogenesis. $n = 20$ cells analysed for all treatment conditions.”

In addition, we have included the data we used to determine the optimal concentration of cephalalexin (in the new Supplementary Fig. 4) necessary to induce cell elongation in *C. crescentus*, as well as the growth curves used to determine appropriate sub-inhibitory concentrations of various compounds (in the new Supplementary Fig. 3).

“Supplementary Fig. 3. Elongation of *C. crescentus* following incubation with cephalexin. *C. crescentus* cells were grown in PYE containing varying concentrations of cephalexin. (a) Samples were taken each hour for a total of three hours, spotted onto an agarose pad, and imaged to observe morphological changes caused by cephalexin exposure. Scale bars = 10 μm . (b) Using MicrobeJ, the cell lengths were extracted and quantified, showing that 3 hours of incubation with 50 $\mu\text{g/mL}$ cephalexin results in a drastically increased average cell length (5.22 μm), almost double that of cells grown in 25 $\mu\text{g/mL}$ cephalexin, which achieved an average cell length of 2.70 μm after two hours of exposure. $N \geq 100$ cells per group.”

And the growth curves with cephalexin -

“Supplementary Fig. 4. *C. crescentus* growth curves under exposure to various compounds. *C. crescentus* *rsaA-467-spytag ΔsapA* grown in the presence of varying concentrations of (a) mitomycin C (MMC), (b) polymyxin B, (c) cephalexin, (d) A22, and (e) chloramphenicol at various concentrations to determine sub-lethal concentrations for pulse-chase labelling experiments. All samples were grown in liquid PYE media at 30 °C, with shaking. (f) *C. crescentus* cells grown in varying concentrations of chloramphenicol over 5 hours. Pulse-chase labelled in SC-mRFP1 (magenta) and SC-sfGFP (green) in PYE (no treatment) or PYE supplemented with 0.5 µg/mL chloramphenicol. Scale bars = 5 µm.”

The goal of our cephalexin-treatment experiments was to disrupt cell division using sub-lethal concentrations of cephalexin; we found this partial inhibition of PBP3 induced cell elongation, a morphological marker which we believe adequately demonstrates that cell development has been significantly disrupted. Arguably, this experiment could be insufficient in linking RsaA secretion to septal PG turnover; however, we believe it demonstrates that the discrete localisation of old and new S-layer is retained despite unsuccessful division. We have amended the text to clarify this-

Line 152: “To test whether local S-layer assembly depends on the cell division machinery, we next treated cells with cephalexin, a cephalosporin antibiotic that inhibits cell division by disrupting peptidoglycan (PG) synthesis at the mid-cell during division(Griffith, 1983, Wang et al., 1998, Costa et al., 2008). We used a concentration of cephalexin that allowed the cells to grow, thus remaining amenable to our pulse-chase labelling methods, but with partial disruption of the cell cycle. Based on growth curves using varying concentrations of cephalexin (Supplementary Fig. 3) and quantification of cephalexin-induced cell elongation (Supplementary Fig. 4), we supplemented the PYE growth medium with 50 µg/mL cephalexin. This sub-lethal level of cephalexin exposure significantly inhibited cell division, and resulted in the formation of lines of connected filamentous cells (Supplementary Fig. 4), consistent with previously published work(Pogliano et al., 1997). These filamentous cells were labelled in the same manner as untreated cells, but with a prolonged chase (3 hours) to allow for growth. Despite cephalexin’s marked impact on cell morphology, cephalexin-treated cells retained a dual-labelled S-layer pattern with distinct regions of old and new S-layer (Fig. 2b and Supplementary Fig. 5). Repeating the co-localisation analysis in these cells confirmed that the old and new S-layer regions were strongly anticorrelated, almost to the same extent as untreated cells (Fig. 2d, average PCC = -0.45, no significant difference in Student’s *t*-test to untreated). The localisation pattern is also consistent between untreated and cephalexin-treated cells, with new S-layer localising to the mid-cell and cell poles (Supplementary Fig. 5a-

d). The cephalixin-treated cells often showed some additional new S-layer in the cell body, equidistant between the cell poles and midcell, at potential sites of disrupted cell division.”

We would also like to point out that now we have additionally used mitomycin C as an alternative cell division inhibitor that causes cellular filamentation and observed similar results (see above and new Fig. 5).

Line 214: “...we treated the cells with sub-lethal concentrations (Supplementary Fig. 3) of mitomycin C (MMC), a DNA-crosslinking agent(Iyer and Szybalski, 1963). MMC induces cell filamentation via the SOS response(Janion, 2008), whereby cells inhibit their own division to prevent the propagation of mutations to progeny cells. This has been well characterised in *C. crescentus*(Modell et al., 2011, Modell et al., 2014) and other bacterial species, including *Streptomyces venezualae*(Falguera et al., 2022, Stratton et al., 2022). We also included a sample containing an increased concentration of cephalixin (100 µg/mL) to ensure the observed retention of S-layer localisation is not concentration-dependent. Both the increased concentration of cephalixin and MMC resulted in cell elongation (Fig. 5a). In addition, both conditions produced cells with an anticorrelated localisation of old and new S-layer, evidenced by strong, negative PCC R-values, comparable to untreated cells (Fig. 5b). This confirmed that inhibiting cell division does not lead to delocalisation of S-layer biogenesis.”

I don't know what the A22 experiments really add. Sure, A22 inhibits MreB and the cells become lemon-shaped because of PG disorganization after MreB is inhibited, but in terms of the model except perhaps adding that PG perturbations in general can influence the polar/midcell disposition of RsaA -Spy, I don't see any value. Except that the A22 inhibition worked, which cannot be said for cephalixin which the authors used in hopes of inhibiting PBP3 (FtsI).

The goal of the experiment was to assess dependence of S-layer biogenesis on the bacterial actin machinery. We found that despite the disrupted localisation of RsaA insertion, the S-layer remained confluent and retained its ultrastructure, demonstrating that the labelling pattern is not a result of failed oligomerisation/binding. This clarification is in the manuscript text –

Line 183: “...we next investigated the effect of disrupting cell elongation and rod morphogenesis by the disruption of the bacterial actin homologue MreB. To do so, we treated *C. crescentus* cells with the compound A22, which has been shown to bind to MreB and disrupt cell shape(van den Ent et al., 2014, Takacs et al., 2010) and cell polarity(Dye et al., 2011, Gitai et al., 2004). We investigated the effect of disruption of MreB filaments on S-layer biogenesis using sub-lethal concentrations of A22 (Supplementary Fig. 3).”

What is the difference between fig 2B and 3A and should fig4 not be embedded in Fig 1?

These were different images from the same experiments. For continuity, the original arrangement of the figure panels was adopted. In the revised manuscript, we have updated

the figures extensively as suggested by the reviewer. We have tried many ways to present a large volume of data, we thank the reviewer for this suggestion.

are the gaps in the RsaA lattice enlarged when cell division is inhibited in FtsZ depletion strain for example?

We attempted to evaluate the effect of direct FtsZ disruption on S-layer biogenesis using the FtsZ inhibitor and stabiliser PC190723 (Andreu et al., 2010, Elsen et al., 2012). Unfortunately, *C. crescentus* appeared completely resistant to the compound and showed no growth or morphological defects (see below).

Growth of *Caulobacter crescentus* under exposure to PC190723.

Given the data presented in the manuscript and the additional experiments conducted following the advice and direction of the reviewers, we would speculate FtsZ depletion would have minimal effect on the localisation of S-layer biogenesis. Cephalixin and mitomycin C (covered earlier in this response) inhibit the progression of the cell cycle by targeting the division site. The size of these gaps is also unlikely to be affected by cell division inhibition, be it through FtsZ depletion or another method, as the S-layer appears completely confluent upon exposure to any cell cycle disrupting compounds.

I don't see the connection between the RsaA lattice gaps and the pole/midcell SpyCatcher labelling experiments. It seems disjointed as two independent observations without causality in this write-up.

Sorry if this was not clear. The connection between the gaps and the fluorescence is shown by Supplementary Fig. 9, the same_sample is being imaged in the light and electron microscope, showing that the sites of S-layer insertion are the sites of lattice gaps

“Supplementary Fig. 5. Cryo-CLEM experiment to confirm that RsaA inserts at disruptions in the cellular S-layer.”

(a) Maximum Z-projection of a Z-stack through a cryo-EM grid containing vitrified, dual-labelled *C. crescentus* cells. Scale bars = 10 μm . Regions of old S-layer are highlighted in magenta (SC-mRFP1), new S-layer in green (SC-sfGFP), and a merge of both channels is provided in the last panel. (b) A zoomed view of the subsection of the micrograph showing the region used for cryo-ET data collection, highlighted in the merged panel of a) (white dashed border), channels are arranged as in a). Scale bars = 5 μm . The inset in the merged channel micrograph shows a re-contrasted image where the layout of the EM grid, and the cell selected for cryo-ET collection is highlighted by a white box. (c) Slice through a tomogram of the dual-labelled *C. crescentus* cell highlighted in b). Components of the cell envelope are labelled using coloured arrows (legend in the bottom left of the panel). A zoomed view of the region highlighted by the red box in panel c), showing a short stalk. Discontinuity in the S-layer can be seen at the stalk tip (likely due to the presence of the holdfast-polysaccharide¹⁰⁵) and at the base of the stalk. Scale bars = 50 nm. A further closeup of the latter is given in the bottom right panel, showing overlapping regions of the S-layer (labelled as “Defect” using a

red arrow). The N-terminal and C-terminal domains (NTD and CTD) of RsaA in the assembled S-layer are marked by white and black arrows, respectively.”

Also, this experiment is followed up by cryo-ET imaging of cell poles and mid-cells where gaps are additionally seen.

“Supplementary Fig. 6. Cryo-ET gallery of cephalixin and A22 treated *C. crescentus* cells. Slices through reconstructed tomograms of (a) cephalixin (50 $\mu\text{g}/\text{mL}$) and (b) A22 (3 $\mu\text{g}/\text{mL}$) treated cells. The left panel in each case shows a view of the cell body (scale bars = 100 nm), while the right panel shows a zoomed-in view of the regions highlighted by the red box. Z-slice has been adjusted in the zoomed views to highlight the cell envelope.”

To summarise, the key point linking the two observations is that same regions of the cell are being imaged in two modalities, namely light and electron microscopy.

Finally it would be very interesting to investigate if SpyCatcher labelling is affected when inhibit LPS synthesis is perturbed with CHIR-090 (LpxC inhibitor) or polymyxin which targets lipidA directly. These inhibitor experiments might work better than the cephalixin (or other beta-lactam) inhibition studies, unless it is done in mutant background that lacks the beta-lactamase or after prior chemical inhibition of the beta-lactamase.

We have performed the experiment with polymyxin-B as suggested. The experiment is presented in the new Fig. 5.

“Fig. 2. Quantification of spatial separation of old and new S-layer following exposure to various compounds.

(a) Pulse-chase labelled *C. crescentus* cells following growth in PYE media with no drug (untreated), or PYE supplemented with one of four compounds: 0.5 $\mu\text{g}/\text{mL}$ mitomycin C; 100 $\mu\text{g}/\text{mL}$; cephalalexin; 7.5 $\mu\text{g}/\text{mL}$ polymyxin B. Scale bar = 5 μm . (b) Co-localisation of mRFP1 and sfGFP fluorescence was quantified using Pearson’s Correlation Coefficient (PCC). Untreated cells and those treated with cell division-inhibiting compounds mitomycin C and cephalalexin showed negative PCC scores (average R-values -0.58, -0.70, -0.45, respectively). Following exposure to the LPS lipid A binding peptide polymyxin B, cells showed positive PCC scores (average R-value 0.84), revealing a disruption of spatially localised S-layer biogenesis. $n = 20$ cells analysed for all treatment conditions.”

Consistent with our expectation, removal or disruption of LPS eradicates RsaA binding to the cell surface due to the loss of the O-antigen anchor (Zik et al., 2022). Following exposure to sub-lethal concentrations of polymyxin B, labelling results showed a loss of spatially separated pulse and chase localisations, as indicated by a positive PCC score. This result confirms earlier findings that LPS is essential for proper S-layer biogenesis (Herdman et al., 2022), also highlighted in the main text -

Line 230: “In stark contrast to inhibiting cell division, spatial separation of both old and new S-layer was also completely disrupted (Fig. 5) when the cells were treated with a sub-lethal concentration (7.5 $\mu\text{g}/\text{mL}$, Supplementary Fig. 3) of polymyxin B (Fig. 5). The activity of polymyxin B is dependent on its binding to lipid A(Morrison and Jacobs, 1976) of LPS. Further,

polymyxin B has been shown to displace cations from the LPS(Jiang et al., 2021, Santos et al., 2017). Disruption of LPS in the outer membrane by polymyxin B led to delocalisation of S-layer biogenesis, consistent with the known dependence of RsaA on LPS for assembly.”

The discussion mentions that these findings have implications on synthetic biology approaches using the S-layer as nanodevice, but is not clear to me how the RsaA pulse-chase localization which still ultimately gives a homogenous distribution of RsaA on the surface, could influence use of the S-layer. Please explain this point!

Many recent examples utilising the *C. crescentus* S-layer as a platform for synthetic biology have been reported (Charrier et al., 2019, Molinari et al., 2022b, Molinari et al., 2021). Fundamental knowledge about the mechanisms of S-layer biogenesis will help in harnessing the power of these systems for synthetic approaches, now discussed in the manuscript -

Line 379: “Moreover, our studies have implications in the design and utilisation of S-layers as platforms for synthetic biology applications¹⁰⁰. The *C. crescentus* S-layer has already been adapted as an engineering hub for purposes ranging from protein display⁹³ to customisable scaffolds for the design of living materials^{44,101}. The exploitation of natural systems for the development of engineered living materials has broad applications¹⁰². In particular, the self-organising and self-healing nature of S-layers provides an ideal scaffold for the design of a high-density biological interaction^{44,101}. We expect that uncovering the mechanisms of spatiotemporal localisation of S-layer biogenesis will help further harness the *C. crescentus*’ S-layer as a platform for synthetic biology. Particularly, our demonstration of the consistent and predictable nature of RsaA integration into the expanding S-layer, supports fine tuning of protein scaffolding on the cell.”

Reviewer #2 (Remarks to the Author):

Herdman et al. report on the localisation of S-layer assembly in the model bacterium *Caulobacter crescentus* and its coordination with cell envelope expansion. To do so, the authors employ pulse-chase fluorescence labelling of a Spy-tagged RsaA derivative, and follow de novo S-layer synthesis using light microscopy and cryoET. This is done both under physiological conditions as well as in conditions where cell division or MreB-coordinated cell elongation are specifically inhibited.

The study finds that S-layer assembly occurs at sites of cell envelope expansion, where it follows on de novo peptidoglycan synthesis.

The observation of localised S-layer expansion at sites of cell growth and division is in itself not new. Comerchi et al. Nature Communications volume 10, 2731 (2019), used a very similar pulse-chase fluorescence labelling protocol, as well as the A22 and Cephalexin-inhibited cell elongation or division, respectively. This current study differs in also labelling de novo peptidoglycan synthesis (i.e. using HADA), and in obtaining a molecular resolution view of the S-layer expansion sites by cryoET. The latter illustrates the presence of defect or dislocation zones from which the S-layer is likely to expand.

The work is well executed and results are beautifully documented.

Thank you for these encouraging comments.

Whilst the cryoET imaging and the HADA labelling do provide a very detailed illustration of the S-layer expansion sites, the fundamental advance of this study over Comerchi et al. 2019 remains limited.

In response to the reviewers' comments, we have further bolstered the manuscript with more experiments, analyses, and text. The advance provided by this study is the ultrastructural investigations at the level of single S-layer hexamers, in addition to testing the cell cycle dependence of S-layer biogenesis (both aspects not covered by Comerchi *et al*). We have clarified this in the text now -

Line 81: "In this study, we have investigated the cell cycle dependence of S-layer biogenesis in *C. crescentus*, using fluorescent microscopy and electron cryotomography (cryo-ET) of cells, which allowed us to visualise the S-layer arrangement on cells, up to the level of single S-layer hexamers. Our results show that S-layer biogenesis is tightly linked with cell elongation and cell growth. We provide evidence showing that cell division and cell envelope biogenesis are regulated at multiple levels,..."

The authors note that the site of RsaA secretion remains unknown. It would be interesting to try resolve this by expression of a fluorescent RsaA fusion protein, and to see if secretion is in any way coordinated with cell expansion. I.e. would the use of A22 result in the build-up of RsaA monomers trapped underneath the S-layer?

We have tried extremely hard to address this question by tagging the RsaDEF secretion machinery (see also our response to reviewer 1 above). Unfortunately, in the time frame of this revision, we cannot provide a clear-cut answer, despite our efforts. We hope to answer this in the next few years.

Localisation of mNeonGreen-RsaD (mNG-RsaD) in *C. crescentus*.

(A) Micrographs of *C. crescentus* cells, pulse labelled in SpyCatcher003 (non-fluorescent) and chased using SC-mRFP1. mNG-RsaD shows a heterogenous distribution of green punctae across the cell. Scale bar = 10 μ m. (B) Some cells show co-localisation of SC-mRFP1 (representing new S-layer) and mNG-RsaD at cell poles. Scale bar = 1 μ m. However, most punctae along the cell body have no associated SC-mRFP1 fluorescence signal. (C) Cryo-ET reveals *C. crescentus* cells expressing mNG-RsaD have a patchy S-layer, absent over much of the cell surface. Scale bar = 500 nm.

Is this what the authors are illustrating in Fig. 3E, i.e. the build-up and eventually crystallisation of RsaA underneath the S-layer? The text and legend are not very clear on this.

This is a misunderstanding, sorry for the confusion. We agree that the site of RsaA secretion is of great interest and plan to explore this further in the future. To address the proposed hypothesis, we would expect a degree of build-up of RsaA in the cytoplasm in the event the secretion machinery is disrupted – however, RsaA appears to be under very tight transcriptional regulation by *Caulobacter*, and any build-up is likely to result in net reduction of RsaA production (Lau et al., 2010, Toporowski et al., 2004, Awram and Smit, 1998). We have updated the text to clarify this further -

Line 75: “RsaA has been suggested to account for between 10-31% of total protein content of the cell(Lau et al., 2010, Awram and Smit, 1998) and appears to be tightly regulated to prevent cytoplasmic build-up of excess protein(Overton et al., 2016, Lau et al., 2010, Toporowski et al., 2004, Awram and Smit, 1998).”

and

Line 206: “One set had severe cellular disruption including invaginated membranes (**Error! Reference source not found.**d and Supplementary Fig. 6); these membrane distortions are likely responsible for the patchy fluorescence and disrupted S-layer localisation.”

Furthermore, now with the structure in hand, did the authors consider using an assembly incompetent mutant of RsaA, or just the cell-anchoring domain to see if these would compete with the sites of S-layer expansion.

0.1 μM RsaA_{NTD}

Addition of RsaA_{NTD} during chase labelling results in new regions of S-layer insertion along the cell body.

C. crescentus cells pulse labelled in SC-mRFP1 (magenta), followed by chase labelling with SC-sfGFP (green) in the presence of 0.1 μM RsaA_{NTD}. Cells retain anticorrelation between old and new regions of S-layer but show additional regions of new S-layer along the cell body (white arrows). This differs from the restriction of new S-layer insertion to the cell pole and midcell (which is still observed in a majority of cells). Scale bars = 1 μm .

As suggested, we tried this experiment. Although the fluorescent labelling was altered, with new regions of RsaA insertion across the cell body, there still evidence of anti-correlation between the old and new S-layer. We interpret this as some RsaA-NTD (cell anchoring domain) molecules competing for and reducing the available sites for tagging (see below). However, we do not think that this experiment offers insights worthy of inclusion into the manuscript.

We want to point out that a similar experiment using full-length RsaA has been done by *Comerci et al*, because their strategy for labelling used exogenous protein.

I.e. is it the availability of free LPS sites or the availability of a sterically free crystals growth edge that become limiting to expansion of the S-layer?

We have conducted an experiment whereby cells are treated with polymyxin B during labelling. The results, as summarised in our new Fig. 5, confirm that binding of the LPS by RsaA can be obstructed by polymyxin B, resulting in delocalisation S-layer biogenesis.

“Fig. 3. Quantification of spatial separation of old and new S-layer following exposure to various compounds.

(a) Pulse-chase labelled *C. crescentus* cells following growth in PYE media with no drug (untreated), or PYE supplemented with one of four compounds: 0.5 $\mu\text{g}/\text{mL}$ mitomycin C; 100 $\mu\text{g}/\text{mL}$; cephalexin; 7.5 $\mu\text{g}/\text{mL}$ polymyxin B. Scale bar = 5 μm . (b) Co-localisation of mRFP1 and sfGFP fluorescence was quantified using Pearson’s Correlation Coefficient (PCC). Untreated cells and those treated with cell division-inhibiting compounds mitomycin C and cephalexin showed negative PCC scores (average R-values -0.58, -0.70, -0.45, respectively). Following exposure to the LPS lipid A binding peptide polymyxin B, cells showed positive PCC scores (average R-value 0.84), revealing a disruption of spatially localised S-layer biogenesis. $n = 20$ cells analysed for all treatment conditions.”

Minor points.

In line 210 and later that paragraph lines 218-19 the authors describe the presence of ‘gaps’ in the S-layer. The images appear to show defects or dislocation of the S-layer growth edge more than showing gaps, so that this terminology may be somewhat misleading.

We agree this terminology is inaccurate and the text has been adapted accordingly.

Line 483: “Components of the cell enveloped have been labelled. Specifically, yellow arrows denote regions of discontinuities or overlaps in the S-layer lattice.”

and

Line 28: “Finally, correlated cryo-light microscopy and electron cryotomographic analysis of regions of S-layer insertion showed the presence of discontinuities in the hexagonal S-layer lattice, contrasting with other S-layers completed by defined symmetric defects”

Reviewer #3 (Remarks to the Author):

This manuscript by Herdman et al describes a series of light microscopy and cryo-tomography experiments focused on the assembly of new S-layer on the surface of *Caulobacter crescentus*. The experiments are very well-designed, the quality of the data is high and the interpretation is reasonable and supported by the presented data. Overall there's very little to criticise here and I commend the authors on an elegantly designed study and well-written manuscript.

Thank you for these encouraging comments.

The work presented here is largely in agreement with previous studies of S-layer growth in multiple species. These are cited in the introduction, but the parallels are largely ignored in the discussion unfortunately.

Sorry about that, we did not want to ignore previous literature. We have therefore, significantly expanded the discussion to correct this -

Line 355: "Interestingly, in *C. difficile*, the secretion machinery of the major structural S-layer protein SlpA, called SecA2, was localised using a modified, fluorescent version of SlpA blocked by secretion by an N-terminally linked dihydrofolate reductase (Oatley et al., 2020). This study found that while SecA2 was localised across the cell surface, SlpA integrated into the growing S-layer at regions of cell wall turnover. These findings are consistent with our proposed model for S-layer biogenesis, despite *C. crescentus* and *C. difficile* having strikingly different cell cycles and envelope organisations (diderm versus monoderm, respectively). In contrast, the S-layer assembly machinery of *H. volcanii* appears to be consistently colocalised with regions of new S-layer biogenesis (Abdul-Halim et al., 2020), also colocalised with FtsZ. This difference in spatial localisation of secretion with biogenesis may arise from the fact that in *H. volcanii*, the S-layer glycoprotein is lipidated, forming an integral part of the membrane. For the *C. crescentus* S-layer, future studies on the localisation of RsaDEF would provide further context into the mechanisms of S-layer secretion, and how this compares to other species."

It would be useful to more fully place these new *Caulobacter* findings in the context of what is already known. I think this is particularly important in light of the observation of boundary defects and gaps at the site of insertion. The authors point out that this has not been observed previously but might this reflect the resolution of the methods used in previous studies? Is it possible that this is a more common feature of bacterial S-layer growth?

We agree with the reviewer's observation that our findings could be more adequately placed among the current literature on the *Caulobacter* S-layer. The text has been updated to address this concern, as well as highlighting the resolution of our cryo-ET imaging -

Line 82: "...using cellular fluorescence microscopy and electron cryotomography (cryo-ET), which allowed us to visualise the S-layer arrangement on cells, down to the level of single S-layer hexamers."

and

Line 335: “Owing to the crystalline nature of the *C. crescentus* S-layer, SLPs (RsaA molecules) are likely able to self-integrate themselves into the growing lattice at gaps in the two-dimensional crystal. This has been observed *in vitro*, where purified RsaA in the presence of calcium spontaneously forms hexameric lattices comparable to those seen on the cell surface (Allievi et al., 2014, Herrmann et al., 2020, Bharat et al.). This presents an ingenious solution to a difficult logistical problem of the polymerisation of the highly ordered S-layer, as it requires no further energetic input from the cell beyond secretion of the constituent SLP into the extracellular milieu, where it will bind the surface and oligomerise. It is remarkable therefore, that other studied S-layers with a similar mid-cell insertion phenotype, do not possess extensive gaps in the lattice, but rather complete the S-layer with defined geometric defects (Deatherage et al., 1983, Messner et al., 1986, Pum et al., 1991). For example, in the archaeal S-layer of *Haloferax volcanii*, pentamers and heptamers were observed on cells, which were coated to near-perfect continuity by the hexagonal S-layer (von Kügelgen et al., 2021). Geometrically, to close a hexagonal sheet, defects or gaps must be present, therefore it will be intriguing to study why different organisms have adopted different solutions to this problem. Future research in this direction will illuminate our understanding of curved lattices in cells, which are ubiquitous across domains of life.”

Also, given the authors previous description of a pool of free subunits underlying the S-layer can you speculate about what these defects tell us about the kinetics of assembly? Why are these gaps not efficiently filled as the underlying cell envelope grows?

A model with a pool of free SLP subunits between the outer membrane and S-layer is consistent with previous reports (Comerci et al., 2019). A major limiting factor in the ability of RsaA to effectively plug gaps in the S-layer could indeed be kinetic, although we cannot be certain. We have added a sentence to the discussion as suggested -

Line 302: “The insertion of new LPS into the OM is probably kinetically faster than the diffusion of RsaA to the O-antigen tip, because discontinuity in the S-layer is seen at these regions.”

In other species distinct grain boundaries have been observed in the S-layer and it is thought that these form as a consequence of S-layer growth. Are there symmetry mismatches observed across the gaps observed here? Is lattice orientation consistent across the *Caulobacter* surface? If so that would seem to be an important point to highlight.

We have looked carefully through our entire dataset, and found one example where there might be such a grain boundary (see below) –

However, we cannot be sure where this is due to a slight change in local curvature of the sheet, therefore we did not include this into the manuscript.

Minor issues:

We agree with all the suggested text corrections and have amended the manuscript accordingly, detailed below.

Line 43: abundant is an odd choice of word here

Has been changed to –

Line43: “There is increasing evidence suggesting that S-layers are prevalent in prokaryotes,..”

Line 98: different terminology used to describe the cell types and in the figure here (swarmer vs non-dividing) – would be helpful to the non-specialist reader to define these more clearly and use a single nomenclature.

Has been clarified now –

Line 112: “To further quantify the relationship between the cell cycle stage and the labelling pattern, the fluorescence profiles of non-dividing swarmer cells (cell length <2 μm) and dividing cells (assigned by the presence of a mid-cell invagination)...”

In addition to the above changes, we have expanded on our explanation of the asymmetric nature of *C. crescentus* cell development in the text. We hope this will make the context of our study clearer to readers.

Line 128: “The *C. crescentus* stalk, which is found on the pole of the dividing cell, is also encompassed by an S-layer(Smit et al., 1992, Bharat et al., 2017). Stalk biogenesis is driven by the recruitment of cell wall biosynthetic machinery at the old pole (Yakhnina and Gitai, 2013, Billini et al., 2019, Kühn et al., 2010), representing a rapid transition from swarmer to sessile cell type in *C. crescentus*, the latter representing the dividing population(Biondi et al., 2006, Kang and Shapiro, 1994, Kühn et al., 2010).”

Line 228: typo “at the several areas”

Has been corrected –

Line 286: “At several areas, two separate two-dimensional sheets of S-layers appear to meet, showing up as a line defect on the cell surface.”

Line 259: typo “the biogenesis different”

Has been corrected –

Line 320: “There is clear tendency for many bacterial and archaeal cells to synchronise the biogenesis of different envelope components...”

REFERENCES

- ABDUL-HALIM, M. F., SCHULZE, S., DILUCIDO, A., PFEIFFER, F., BISSON FILHO, A. W. & POHLSCHRODER, M. 2020. Lipid anchoring of archaeosortase substrates and midcell growth in haloarchaea. *MBio*, 11, e00349-20.
- ALLIEVI, M. C., PALOMINO, M. M., PRADO ACOSTA, M., LANATI, L., RUZAL, S. M. & SANCHEZ-RIVAS, C. 2014. Contribution of S-layer proteins to the mosquitocidal activity of *Lysinibacillus sphaericus*. *PLoS One*, 9, e111114.
- ANDREU, J. M., SCHAFFNER-BARBERO, C., HUECAS, S., ALONSO, D., LOPEZ-RODRIGUEZ, M. L., RUIZ-AVILA, L. B., NÚÑEZ-RAMÍREZ, R., LLORCA, O. & MARTÍN-GALIANO, A. J. 2010. The antibacterial cell division inhibitor PC190723 is an FtsZ polymer-stabilizing agent that induces filament assembly and condensation. *Journal of Biological Chemistry*, 285, 14239-14246.
- AWRAM, P. & SMIT, J. 1998. The *Caulobacter crescentus* paracrystalline S-layer protein is secreted by an ABC transporter (type I) secretion apparatus. *J Bacteriol*, 180, 3062-9.
- BEN-SASSON, A. J., WATSON, J. L., SHEFFLER, W., JOHNSON, M. C., BITTLESTON, A., SOMASUNDARAM, L., DECARREAU, J., JIAO, F., CHEN, J., MELA, I., DRABEK, A. A., JARRETT, S. M., BLACKLOW, S. C., KAMINSKI, C. F., HURA, G. L., DE YOREO, J. J., KOLLMAN, J. M., RUOHOLA-BAKER, H., DERIVERY, E. & BAKER, D. 2021. Design of biologically active binary protein 2D materials. *Nature*, 589, 468-473.
- BHARAT, T. A. M., KUREISAITE-CIZIENE, D., HARDY, G. G., YU, E. W., DEVANT, J. M., HAGEN, W. J. H., BRUN, Y. V., BRIGGS, J. A. G. & LÖWE, J. 2017. Structure of the hexagonal surface layer on *Caulobacter crescentus* cells. *Nat Microbiol*, 2, 17059.
- BILLINI, M., BIBOY, J., KÜHN, J., VOLLMER, W. & THANBICHLER, M. 2019. A specialized MreB-dependent cell wall biosynthetic complex mediates the formation of stalk-specific peptidoglycan in *Caulobacter crescentus*. *PLOS Genetics*, 15, e1007897.
- BINGLE, W. H., NOMEILLINI, J. F. & SMIT, J. 1997. Linker mutagenesis of the *Caulobacter crescentus* S-layer protein: toward a definition of an N-terminal anchoring region and a C-terminal secretion signal and the potential for heterologous protein secretion. *J Bacteriol*, 179, 601-11.
- BIONDI, E. G., SKERKER, J. M., ARIF, M., PRASOL, M. S., PERCHUK, B. S. & LAUB, M. T. 2006. A phosphorelay system controls stalk biogenesis during cell cycle progression in *Caulobacter crescentus*. *Mol Microbiol*, 59, 386-401.
- CHARRIER, M., LI, D., MANN, V. R., YUN, L., JANI, S., RAD, B., COHEN, B. E., ASHBY, P. D., RYAN, K. R. & AJO-FRANKLIN, C. M. 2019. Engineering the S-Layer of *Caulobacter crescentus* as a Foundation for Stable, High-Density, 2D Living Materials. *ACS Synth Biol*, 8, 181-190.
- COMERCI, C. J., HERRMANN, J., YOON, J., JABBARPOUR, F., ZHOU, X., NOMEILLINI, J. F., SMIT, J., SHAPIRO, L., WAKATSUKI, S. & MOERNER, W. E. 2019. Topologically-guided

- continuous protein crystallization controls bacterial surface layer self-assembly. *Nat Commun*, 10, 2731.
- COSTA, T., PRIYADARSHINI, R. & JACOBS-WAGNER, C. 2008. Localization of PBP3 in *Caulobacter crescentus* is highly dynamic and largely relies on its functional transpeptidase domain. *Mol Microbiol*, 70, 634-51.
- DEATHERAGE, J. F., TAYLOR, K. A. & AMOS, L. A. 1983. Three-dimensional arrangement of the cell wall protein of *Sulfolobus acidocaldarius*. *J Mol Biol*, 167, 823-48.
- DUVAL, M., LEWIS, C. J., NOMELLINI, J. F., HORWITZ, M. S., SMIT, J. & CAVACINI, L. A. 2011. Enhanced Neutralization of HIV by Antibodies Displayed on the S-Layer of *Caulobacter crescentus*. *Antimicrobial Agents and Chemotherapy*, 55, 5547-5552.
- DYE, N. A., PINCUS, Z., FISHER, I. C., SHAPIRO, L. & THERIOT, J. A. 2011. Mutations in the nucleotide binding pocket of MreB can alter cell curvature and polar morphology in *Caulobacter*. *Molecular microbiology*, 81, 368-394.
- ELSEN, N. L., LU, J., PARTHASARATHY, G., REID, J. C., SHARMA, S., SOISSON, S. M. & LUMB, K. J. 2012. Mechanism of action of the cell-division inhibitor PC190723: modulation of FtsZ assembly cooperativity. *Journal of the American Chemical Society*, 134, 12342-12345.
- FALGUERA, J. V. T., STRATTON, K. J., BUSH, M. J., JANI, C., FINDLAY, K. C., SCHLIMPERT, S. & NODWELL, J. R. 2022. DNA damage-induced block of sporulation in *Streptomyces venezuelae* involves downregulation of *ssgB*. *Microbiology*, 168.
- GITAI, Z., DYE, N. & SHAPIRO, L. 2004. An actin-like gene can determine cell polarity in bacteria. *Proceedings of the National Academy of Sciences of the United States of America*, 101, 8643-8648.
- GRIFFITH, R. 1983. The pharmacology of cephalexin. *Postgraduate medical journal*, 59, 16-27.
- HERDMAN, M., VON KÜGELGEN, A., KUREISAITE-CIZIENE, D., DUMAN, R., EL OMARI, K., GARMAN, E. F., KJAER, A., KOLOKOURIS, D., LÖWE, J., WAGNER, A., STANSFELD, P. J. & BHARAT, T. A. M. 2022. High-resolution mapping of metal ions reveals principles of surface layer assembly in *Caulobacter crescentus* cells. *Structure*, 30, 215-228 e5.
- HERRMANN, J., LI, P. N., JABBARPOUR, F., CHAN, A. C. K., RAJKOVIC, I., MATSUI, T., SHAPIRO, L., SMIT, J., WEISS, T. M., MURPHY, M. E. P. & WAKATSUKI, S. 2020. A bacterial surface layer protein exploits multistep crystallization for rapid self-assembly. *Proc Natl Acad Sci U S A*, 117, 388-394.
- IYER, V. & SZYBALSKI, W. 1963. A molecular mechanism of mitomycin action: linking of complementary DNA strands. *Proceedings of the National Academy of Sciences*, 50, 355-362.
- JANION, C. 2008. Inducible SOS response system of DNA repair and mutagenesis in *Escherichia coli*. *International journal of biological sciences*, 4, 338.

- JIANG, X., SUN, Y., YANG, K., YUAN, B., VELKOV, T., WANG, L. & LI, J. 2021. Coarse-grained simulations uncover Gram-negative bacterial defense against polymyxins by the outer membrane. *Computational and Structural Biotechnology Journal*, 19, 3885-3891.
- KANG, P. J. & SHAPIRO, L. 1994. Cell cycle arrest of a *Caulobacter crescentus* *secA* mutant. *Journal of Bacteriology*, 176, 4958-4965.
- KÜHN, J., BRIEGEL, A., MÖRSCHER, E., KAHNT, J., LESER, K., WICK, S., JENSEN, G. J. & THANBICHLER, M. 2010. Bactofilins, a ubiquitous class of cytoskeletal proteins mediating polar localization of a cell wall synthase in *Caulobacter crescentus*. *The EMBO Journal*, 29, 327-339.
- LAU, J. H. Y., NOMEILLINI, J. F. & SMIT, J. 2010. Analysis of high-level S-layer protein secretion in *Caulobacter crescentus*. *Canadian Journal of Microbiology*, 56, 501-514.
- MESSNER, P., PUM, D., SARA, M., STETTER, K. O. & SLEYTR, U. B. 1986. Ultrastructure of the cell envelope of the archaebacteria *Thermoproteus tenax* and *Thermoproteus neutrophilus*. *J Bacteriol*, 166, 1046-54.
- MODELL, J. W., HOPKINS, A. C. & LAUB, M. T. 2011. A DNA damage checkpoint in *Caulobacter crescentus* inhibits cell division through a direct interaction with FtsW. *Genes Dev*, 25, 1328-43.
- MODELL, J. W., KAMBARA, T. K., PERCHUK, B. S. & LAUB, M. T. 2014. A DNA Damage-Induced, SOS-Independent Checkpoint Regulates Cell Division in *Caulobacter crescentus*. *PLOS Biology*, 12, e1001977.
- MOLINARI, S., TESORIERO, R. F. & AJO-FRANKLIN, C. M. 2021. Bottom-up approaches to engineered living materials: Challenges and future directions. *Matter*, 4, 3095-3120.
- MOLINARI, S., TESORIERO, R. F., JR., LI, D., SRIDHAR, S., CAI, R., SOMAN, J., RYAN, K. R., ASHBY, P. D. & AJO-FRANKLIN, C. M. 2022a. A de novo matrix for macroscopic living materials from bacteria. *Nat Commun*, 13, 5544.
- MOLINARI, S., TESORIERO, R. F., LI, D., SRIDHAR, S., CAI, R., SOMAN, J., RYAN, K. R., ASHBY, P. D. & AJO-FRANKLIN, C. M. 2022b. A de novo matrix for macroscopic living materials from bacteria. *Nature Communications*, 13, 5544.
- MORRISON, D. C. & JACOBS, D. M. 1976. Binding of polymyxin B to the lipid A portion of bacterial lipopolysaccharides. *Immunochemistry*, 13, 813-818.
- NGUYEN, P. Q., COURCHESNE, N. M. D., DURAJ-THATTE, A., PRAVESCHOTINUNT, P. & JOSHI, N. S. 2018. Engineered living materials: prospects and challenges for using biological systems to direct the assembly of smart materials. *Advanced Materials*, 30, 1704847.
- NOMEILLINI, J. F., DUNCAN, G., DOROCICZ, I. R. & SMIT, J. 2007. S-layer-mediated display of the immunoglobulin G-binding domain of streptococcal protein G on the surface of *Caulobacter crescentus*: development of an immunoactive reagent. *Appl Environ Microbiol*, 73, 3245-53.

- OATLEY, P., KIRK, J. A., MA, S., JONES, S. & FAGAN, R. P. 2020. Spatial organization of *Clostridium difficile* S-layer biogenesis. *Sci Rep*, 10, 14089.
- VERTON, K. W., PARK, D. M., YUNG, M. C., DOHNALKOVA, A. C., SMIT, J. & JIAO, Y. 2016. Two Outer Membrane Proteins Contribute to *Caulobacter crescentus* Cellular Fitness by Preventing Intracellular S-Layer Protein Accumulation. *Appl Environ Microbiol*, 82, 6961-6972.
- POGLIANO, J., POGLIANO, K., WEISS, D. S., LOSICK, R. & BECKWITH, J. 1997. Inactivation of FtsI inhibits constriction of the FtsZ cytokinetic ring and delays the assembly of FtsZ rings at potential division sites. *Proceedings of the National Academy of Sciences of the United States of America*, 94, 559-564.
- PUM, D., MESSNER, P. & SLEYTR, U. B. 1991. Role of the S layer in morphogenesis and cell division of the archaeobacterium *Methanococcus sinense*. *J Bacteriol*, 173, 6865-73.
- SANTOS, D. E. S., POL-FACHIN, L., LINS, R. D. & SOARES, T. A. 2017. Polymyxin Binding to the Bacterial Outer Membrane Reveals Cation Displacement and Increasing Membrane Curvature in Susceptible but Not in Resistant Lipopolysaccharide Chemotypes. *Journal of Chemical Information and Modeling*, 57, 2181-2193.
- SMIT, J., ENGELHARDT, H., VOLKER, S., SMITH, S. H. & BAUMEISTER, W. 1992. The S-layer of *Caulobacter crescentus*: three-dimensional image reconstruction and structure analysis by electron microscopy. *J Bacteriol*, 174, 6527-38.
- STRATTON, K. J., BUSH, M. J., CHANDRA, G., STEVENSON, C. E. M., FINDLAY, K. C. & SCHLIMPERT, S. 2022. Genome-Wide Identification of the LexA-Mediated DNA Damage Response in *Streptomyces venezuelae*. *Journal of Bacteriology*, 204, e00108-22.
- SULKOWSKI, N. I., HARDY, G. G., BRUN, Y. V. & BHARAT, T. A. M. 2019. A Multiprotein Complex Anchors Adhesive Holdfast at the Outer Membrane of *Caulobacter crescentus*. *J Bacteriol*, 201.
- TAKACS, C. N., POGGIO, S., CHARBON, G., PUCHEAULT, M., VOLLMER, W. & JACOBS-WAGNER, C. 2010. MreB drives de novo rod morphogenesis in *Caulobacter crescentus* via remodeling of the cell wall. *J Bacteriol*, 192, 1671-84.
- TOPOROWSKI, M. C., NOMELLINI, J. F., AWRAM, P. & SMIT, J. 2004. Two outer membrane proteins are required for maximal type I secretion of the *Caulobacter crescentus* S-layer protein. *J Bacteriol*, 186, 8000-9.
- VAN DEN ENT, F., IZORÉ, T., BHARAT, T. A. M., JOHNSON, C. M. & LÖWE, J. 2014. Bacterial actin MreB forms antiparallel double filaments. *eLife*, 3, e02634.
- VON KÜGELGEN, A., ALVA, V. & BHARAT, T. A. M. 2021. Complete atomic structure of a native archaeal cell surface. *Cell Rep*, 37, 110052.

- WANG, L., KHATTAR, M. K., DONACHIE, W. D. & LUTKENHAUS, J. 1998. FtsI and FtsW are localized to the septum in *Escherichia coli*. *Journal of bacteriology*, 180, 2810-2816.
- WERNER, J. N., CHEN, E. Y., GUBERMAN, J. M., ZIPPILLI, A. R., IRGON, J. J. & GITAI, Z. 2009. Quantitative genome-scale analysis of protein localization in an asymmetric bacterium. *Proceedings of the National Academy of Sciences*, 106, 7858-7863.
- YAKHNINA, A. A. & GITAI, Z. 2013. Diverse functions for six glycosyltransferases in *Caulobacter crescentus* cell wall assembly. *Journal of bacteriology*, 195, 4527-4535.
- ZIK, J. J., YOON, S. H., GUAN, Z., STANKEVICIUTE SKIDMORE, G., GUDDOOR, R. R., DAVIES, K. M., DEUTSCHBAUER, A. M., GOODLETT, D. R., KLEIN, E. A. & RYAN, K. R. 2022. *Caulobacter* lipid A is conditionally dispensable in the absence of fur and in the presence of anionic sphingolipids. *Cell Reports*, 39, 110888.

Reviewer #1 (Remarks to the Author):

The manuscript describes chemical/antibiotic perturbation to infer something about the mechanism of new S-layer subunits at the division plane and the cell poles.

This approach is ok as a first approximation, but it does not suffice as only means, without genetic manipulation using depletion or dominant negative approaches, to make robust conclusions about the underlying mechanisms, especially in an organism where genetic manipulation is easy and in which, conversely, antibiotic action and targeting is poorly documented.

Just because one beta-lactam happens to target PBP3 in *E. coli*, does not mean that the same antibiotic will also inhibit PBP3 in *Caulobacter* for example. Consider aztreonam for example: this is by far the best beta-lactam to inhibit PBP3 in *E. coli*. It is ineffective in *Caulobacter*, even when the *Caulobacter* metallo-beta-lactamase is knocked out.

I have criticized the cephalixin experiments before. *Caulobacter*'s metallo-beta-lactamase that is very active destroys most beta-lactam antibiotics including cephalixin (but, paradoxically, it does not cleave aztreonam, see above). Because of this problem, researchers typically use very high concentrations to inhibit septal synthesis to swamp the beta-lactamase with substrate, but still the inhibition by beta-lactams does not work well and at this concentration other PBPs other than the septal PBP3 could be targeted by the antibiotic that at low concentrations could be specific for the septal PBP. Such off-target PBP action would create other cell wall defects.

Then, as shown here, cephalixin treatment does not give rise to smooth cells blocked in division. So how can one infer anything about the dependence of RsaA insertion on the divisome? Moreover, even if one would treat cells lacking the beta-lactamase to obtain efficient filamentation by blocking PBP3 with a beta-lactam, it is still clear that no conclusion can be drawn on the earliest and defining event on cell division, FtsZ recruitment to the division site. So, clearly the question of whether RsaA insertion at midcell is FtsZ-dependent has not been addressed with the cephalixin experiments. It only shows (but with reasonable doubt) that some PBPs are dispensable for RsaA insertion at midcell, but nothing can be concluded about FtsZ without depletion, or a dominant negative or expression of a division inhibitor. So the conclusion that RsaA localization does not depend on the division is simply not proven and a false claim.

The same logic applies to MreB inhibition with A22. A22 can target other proteins as well other than MreB. In this case however, A22 treatment works well and gives the expected phenotype, however, to eliminate off-target effects, the authors should confirm that the A22-resistant MreB mutant no longer shows the RsaA delocalization. Of course, the best experiment would be to deplete WT MreB or inhibit it by overexpression of a dominant negative or by overexpression MreB-inhibitor discovered in the Gitai lab.

It also feels the same way about the polymyxin inhibition experiments to conclude anything about how RsaA binds the O-antigen. In *Caulobacter*, the lipid A component of LPS does not contain the phosphates that are typically important for efficient inhibition with polymyxin (compared to *E. coli*, see the papers from Jon Smit, PMID: 18387917). So how does it function in *Caulobacter*? Does it still bind lipidA or just displace the divalent cations? And anyhow, how does polymyxin treatment which will target lipidA affect the O-antigen chain that RsaA will bind to? Is the idea that polymyxin targeting of lipidA leads to loss of all O-antigen so that no RsaA signal remains at the pole when spycatcher is added? Are the authors suggesting that there is no O-antigen at this site? If understood the model correctly RsaA will only bind the antigen, so if there is no O-antigen in the middle then RsaA can no longer bind there, but is there O-antigen still elsewhere? Or will polymyxin lead to a complete loss of O-antigen and can this be shown? This is not well explained and again there are no genetic experiments to support the unknown and unclear mechanism about the lipidA or O-antigen dependence for midcell localization of RsaA. Again, in this case I find that chemical perturbation provides some very good hints, but it does not suffice to make a firm statement about the mechanism. Does inhibition of lipid A synthesis with CHIR-90 (targets LpxC) give the same effect?

I don't understand why the mitomycinC (MMC) experiments are described with the polymyxin section. The argument is that MMC induces SOS and there is a division inhibitor that is induced by SOS, BUT this division inhibitor does not act at the level of FtsZ. It inhibits FtsW and again the

question is whether MMC treatment suffices to make a statement about affecting septal cell wall synthesis, but it certainly does NOT resolve the question of whether RsaA localization is FtsZ dependent and dependent on the division machine.

I want to reiterate that the finding that RsaA inserts at midcell is clear and compelling and important. But the manuscript still falls short about convincingly demonstrating anything about the spatial regulation of this process.

In summary (as explained above), I find that phrase in the summary "... localized S-layer insertion is unaltered when cell division is inhibited, but depends crucially on bacterial molecules MreB and lipopolysaccharide." is unwarranted without supporting genetic evidence.

Reviewer #2 (Remarks to the Author):

The authors provide extensive additional experiments that bolster the proposed model of S-layer insertion at sites of PG expansion. The cryoET imaging shows S-layer dislocations at the poles or division planes, in agreement with the proposed model that incorporation of new RsaA units are directed to sites with gaps in the S-layer. Final proof could be provided by cryoET of cells where RsaA can be selectively depleted (for example using a RsaA knockout with plasmid-encoded *rsaA* under control of *Para*), which would be expected to show gaps in the S-layer at the sites here labelled by SG-sfGFP. My support of the study is not conditional on performing this experiment, however.

Response to reviewers' comments for Herdman *et al*

Black text – reviewer comment

Red text – author response

Blue text – manuscript excerpt

Reviewer #1 (Remarks to the Author):

The manuscript describes chemical/antibiotic perturbation to infer something about the mechanism of new S-layer subunits at the division plane and the cell poles.

This approach is ok as a first approximation, but it does not suffice as only means, without genetic manipulation using depletion or dominant negative approaches, to make robust conclusions about the underlying mechanisms, especially in an organism where genetic manipulation is easy and in which, conversely, antibiotic action and targeting is poorly documented.

We would like to thank the reviewer for their constructive comments. We agree that genetic manipulation of *C. crescentus* would likely provide great insight into the mechanisms of S-layer biogenesis, and this will be a focus of our research going forward. However, the observations made following incubation of *C. crescentus* with a variety of compounds provide significant insight into S-layer biogenesis, especially with the use of two different but complementary techniques, namely fluorescence microscopy and electron cryotomography.

Nonetheless, we recognise the reviewer's concerns, and thus we have substantially toned down the messaging of the manuscript related to the inhibitors targeting cell division. We fully take on board that we are not directly depleting FtsZ molecules, and this is now reflected in the revised text, detailed below.

Just because one beta-lactam happens to target PBP3 in *E. coli*, does not mean that the same antibiotic will also inhibit PBP3 in *Caulobacter* for example. Consider aztreonam for example: this is by far the best beta-lactam to inhibit PBP3 in *E. coli*. It is ineffective in *Caulobacter*, even when the *Caulobacter* metallo-beta-lactamase is knocked out.

The reviewer is of course correct that *C. crescentus* encodes a metallo- β -lactamase that defends the cell against cephalixin. However, based on the provided OD₆₀₀ growth curves and the clearly quantifiable change to cell lengths, we believe that *C. crescentus* cell division has been suitably compromised for the purposes of our study. Our goal was to perturb cell division and induce cell filamentation, rather than to completely inhibit PBP3, which was achieved using cephalixin. In light of the reviewer's comments, we do realise that all aspects of cell division cannot be arrested with cephalixin, and therefore, we have amended the text to clarify this miscommunication accordingly.

Line 152

"To test how S-layer biogenesis may be affected by disrupted cell division, we next treated cells with cephalixin, a cephalosporin antibiotic that inhibits cell division by inhibiting penicillin-binding protein 3 (PBP3), which is a divisome-associated protein mediating PG synthesis at the mid-cell during division⁵⁶⁻⁵⁸. We used a concentration of cephalixin that allowed the cells to grow, thus remaining amenable to our pulse-chase labelling methods, but with partial disruption of cell division."

I have criticized the cephalixin experiments before. *Caulobacter*'s metallo-beta-lactamase that is very active destroys most beta-lactam antibiotics including cephalixin (but,

paradoxically, it does not cleave aztreonam, see above). Because of this problem, researchers typically use very high concentrations to inhibit septal synthesis to swamp the beta-lactamase with substrate, but still the inhibition by beta-lactams does not work well and at this concentration other PBPs other than the septal PBP3 could be targeted by the antibiotic that at low concentrations could be specific for the septal PBP. Such off-target PBP action would create other cell wall defects.

We agree that there are likely other off-target effects of cephalixin activity; such effects can be seen in our cryo-ET data where we observe intracellular compartmentalisation. However, we would argue that the surprising finding here is that despite using such a high concentration of cephalixin, the anticorrelation of old and new S-layer was retained. In our opinion, this strengthens our conclusion that PBP3 activity and unperturbed cell division is not essential for localised S-layer biogenesis. Nevertheless, we have clearly mentioned the possible off target effects in the results section.

Line 163

“Despite cephalixin’s marked impact on cell morphology, which could be caused by inhibition of other PBPs beyond PBP3, remarkably cephalixin-treated cells retained a dual-labelled S-layer pattern with distinct regions of old and new S-layer (Fig. 3b and Supplementary Fig. 4).”

Then, as shown here, cephalixin treatment does not give rise to smooth cells blocked in division. So how can one infer anything about the dependence of RsaA insertion on the divisome?

The effect of cephalixin is clear and quantified in our reported growth curves, as well as in the change in morphology of the cells. We did not want to completely block division, but rather partially inhibit to allow for our pulse-chase labelling.

Moreover, even if one would treat cells lacking the beta-lactamase to obtain efficient filamentation by blocking PBP3 with a beta-lactam, it is still clear that no conclusion can be drawn on the earliest and defining event on cell division, FtsZ recruitment to the division site. So, clearly the question of whether RsaA insertion at midcell is FtsZ-dependent has not been addressed with the cephalixin experiments. It only shows (but with reasonable doubt) that some PBPs are dispensable for RsaA insertion at midcell, but nothing can be concluded about FtsZ without depletion, or a dominant negative or expression of a division inhibitor. So the conclusion that RsaA localization does not depend on the division is simply not proven and a false claim.

We would like to apologise to the reviewer for the misunderstanding; we are not suggesting that FtsZ drives S-layer biogenesis or recruits RsaA. Our data suggests that PG remodelling and subsequent cell wall expansion drives the separation of the S-layer, allowing new RsaA to integrate into gaps in the lattice. While this form of cell wall remodelling is most often associated with the divisome, it is not dependent upon it. This model explains why S-layer biogenesis also strongly colocalises with the stalk base, another site of major cell wall synthesis and expansion.

We acknowledge that this concern is likely due to our own miscommunication, and so have updated the manuscript to remove misleading mentions of FtsZ. In addition, we have relocated Supplementary Fig. 2. to the main text (as a new Fig. 2.), to make the incorporation of RsaA at the cell stalk more prominent (see below). The figure has been referred to in the main text accordingly, with appropriate modifications. In addition, we have modified Fig. 9

(formerly Fig. 8) to highlight polar localisation of new S-layer and removed reference to FtsZ to avoid confusion.

“Fig. 2. S-layer localisation patterns in dividing and non-dividing *C. crescentus* cells. Comparison of labelling in (a) non-dividing cells and (b) dividing, stalked cells. Cells were synchronised prior to pulse-chase labelling using SC-mRFP1 and SC-sfGFP as described. Polar labelling can be seen in all cells, but mid-cell labelling is only apparent in dividing cells. Additionally, dual-coloured stalks (SC-sfGFP at the base of the stalk, and SC-mRFP1 at the stalk tip) are indicated by an asterisk. This is consistent with previous research that shows that new stalk material is created from the base of the stalk¹⁰⁸. Scale bars = 1 µm.”

and

Line 126

“As a control, *C. crescentus* cell cultures were briefly synchronised using density centrifugation with Percoll and pulse-chase labelled as above, resulting in a similar labelling pattern to non-synchronised cells (Fig. 2). In addition, cells labelled in this manner often displayed variable stalk labelling, including occasional dual-labelled stalks (Fig. 2b). Stalks are polar appendages produced by division-competent *C. crescentus* cells, which are also encompassed by an S-layer^{29,36}. Stalk biogenesis is driven by the recruitment of peptidoglycan (PG) remodelling machinery at the old cell pole⁴⁶⁻⁴⁸, following a rapid transition from swarmer to sessile cell type in *C. crescentus*, the latter representing the dividing population⁴⁸⁻⁵⁰. In dual-labelled cells, new S-layer labelled in SC-sfGFP was localised toward to stalk base, further affirming the potential colocalisation of S-layer biogenesis with underlying cell wall turnover.”

“Fig. 9. Proposed model of cell-expansion dependent S-layer insertion in *C. crescentus* (a) Micrographs showing individual *C. crescentus* cells of varying length, beginning with a non-dividing cell and ending with pre-divisional cell. Top row shows merged SC-mRFP1 and SC-sfGFP signals, bottom row shows HADA signal. Scale bars = 1 μ m. (b) Schematic representation of the relationship between re-localisation of S-layer biogenesis to regions of cell wall turnover and expansion (green arrows). (c) Model of S-layer biogenesis in *C. crescentus*; unfolded, monomeric RsaA is exported from the cytoplasm into the extracellular space, whereupon it binds LPS. RsaA then diffuses along the LPS until it finds a region of surface expansion corresponding to a gap in the S-layer, driven by PG turnover and polymerisation in the cell wall, leading to RsaA binding to the tip of the LPS and oligomerising with the pre-existing S-layer to complete the lattice.”

The same logic applies to MreB inhibition with A22. A22 can target other proteins as well other than MreB. In this case however, A22 treatment works well and gives the expected phenotype, however, to eliminate off-target effects, the authors should confirm that the A22-resistant MreB mutant no longer shows the RsaA delocalization. Of course, the best experiment would be to deplete WT MreB or inhibit it by overexpression of a dominant negative or by overexpression MreB-inhibitor discovered in the Gitai lab.

We agree that given the downstream effects of MreB inhibition, the effects of A22 should be cautiously interpreted. We have highlighted this in the text, to address the reviewer’s concern.

Line 204

“MreB is a major cytoskeletal component of *C. crescentus*, mediating cell elongation, stalk biogenesis, and cell polarity^{47,67,68}.”

and

Line 229

“Dysregulation of MreB should be cautiously interpreted, as several cellular processes are potentially being simultaneously disrupted.”

It also feel the same way about the polymyxin inhibition experiments to conclude anything about how RsaA binds the O-antigen. In *Caulobacter*, the lipid A component of LPS does not

contain the phosphates that are typically important for efficient inhibition with polymyxin (compared to *E. coli*, see the papers from Jon Smit, PMID: 18387917). So how does it function in *Caulobacter*? Does it still bind lipid A or just displace the divalent cations? And anyhow, how does polymyxin treatment which will target lipid A affect the O-antigen chain that RsaA will bind to? Is the idea that polymyxin targeting of lipid A leads to loss of all O-antigen so that no RsaA signal remains at the pole when spycatcher is added? Are the authors suggesting that there is no O-antigen at this site? If understood the model correctly RsaA will only bind the antigen, so if there is no O-antigen in the middle then RsaA can no longer bind there, but is there O-antigen still elsewhere? Or will polymyxin lead to a complete loss of O-antigen and can this be shown? This is not well explained and again there are no genetic experiments to support the unknown and unclear mechanism about the lipid A or O-antigen dependence for midcell localization of RsaA. Again, in this case I find that chemical perturbation provides some very good hints, but it does not suffice to make a firm statement about the mechanism. Does inhibition of lipid A synthesis with CHIR-90 (targets LpxC) give the same effect?

This was an experiment suggested by the reviewer in the first round. While we agree that questions remain, the indication is clear. We believe polymyxin B is most likely creating more gaps in the S-layer, either by damaging the LPS or dissociating cations, leading to additional sites of RsaA integration. We have amended the text to include an expanded explanation of polymyxin B's effects on S-layer biogenesis, to mitigate the reviewer's concerns.

Line 236

"LPS-inhibition by polymyxin-B, but not inhibition of protein synthesis, disrupts discrete S-layer localisation.

Given the dramatically different effects on S-layer localisation following exposure to cell-cycle disrupting compounds, we sought to explore how biogenesis might be affected by targeting aspects of RsaA synthesis and S-layer assembly. The addition of sublethal concentrations of polymyxin B (7.5 µg/mL, Supplementary Fig. 2d), an LPS-binding antibiotic, resulted in a dramatic loss of anticorrelation between old and new regions of S-layer, confirmed by PCC analysis (Fig. 6). While the exact mechanisms by which polymyxin B disrupts localised S-layer biogenesis cannot be inferred, our results show clearly that localised S-layer biogenesis is LPS dependent. The activity of polymyxin B is proposed to depend on its binding to lipid A⁷⁰, leading to disruption. An alternate hypothesis is that polymyxin B displaces cations like Ca²⁺ from the LPS causing inhibition⁷²⁻⁷⁴. The presence of SC-associated fluorescence suggests that RsaA-467-ST, and therefore the O-antigen and lipid A, are all present on our polymyxin-B-treated cells. It is possible that polymyxin B binding to the LPS disrupts cation and RsaA binding, resulting in loss of the S-layer and an increased frequency of gaps, leading to the observed loss of anticorrelation of new and old S-layer regions."

I don't understand why the mitomycinC (MMC) experiments are described with the polymyxin section. The argument is that MMC induces SOS and there is a division inhibitor that is induced by SOS, BUT this division inhibitor does not act at the level of FtsZ. It inhibits FtsW and again the question is whether MMC treatment suffices to make a statement about affecting septal cell wall synthesis, but it certainly does NOT resolve the question of whether RsaA localization is FtsZ dependent and dependent on the division machine.

We agree with the reviewer that in revising our manuscript, the mitomycin-C data has been inappropriately placed. We have updated the manuscript, presenting cephalixin and MMC data together, given their comparable effects on cell morphology.

Line 175

“To ensure the observed retention of discrete S-layer localisation is not concentration-dependent, cells were incubated with an increased concentration of cephalixin (100 µg/mL), which results in arrest of culture growth, as measured by OD₆₀₀ (Supplementary Fig. 2a). In addition, we incorporated the division-disrupting drug mitomycin-C (MMC) into the labelling protocol at sublethal concentrations in an additional experiment (Supplementary Fig. 2b). MMC is a well-characterised DNA-crosslinking agent⁶⁰ that induces cell filamentation via the SOS response⁶¹. Under MMC treatment, cells inhibit their division, preventing the propagation of mutations to progeny cells. This has been well characterised in *C. crescentus*^{62,63} and other bacterial species, including *Streptomyces venezuelae*^{64,65}. As expected, both the increased concentration of cephalixin and MMC resulted in cell elongation (Fig. 6a). In addition, both conditions produced cells with an anticorrelated localisation of old and new S-layer, evidenced by strong, negative PCC R-values, comparable to untreated cells (Fig. 6b). Together, this confirms that disrupting cell division through multiple pathways does not lead to delocalisation of S-layer biogenesis.”

To aid in the presentation of our data and readability of the manuscript, the location experiments using chloramphenicol has also been amended.

Line 253

“In contrast, inhibition of protein synthesis by the addition of chloramphenicol (Supplementary Fig. 2e) appeared to have no observable effect on new and old S-layer anticorrelation, resulting in cells with labelling comparable to that of untreated cells (Supplementary Fig. 2f). Chloramphenicol, which acts by binding the 50S subunit of the bacterial ribosome, inhibits protein synthesis⁷⁶. This inhibitory effect, despite RsaA being the highest copy number protein of the cell, is a nonspecific effect, resulting in a global reduction of protein synthesis. It is unsurprising then that it results in a significant growth defect (Supplementary Fig. 2e) and, while cells are slowed in their growth, they retain their usual labelling pattern of separated regions of new and old S-layers.”

I want to reiterate that the finding that RsaA inserts at midcell is clear and compelling and important. But the manuscript still falls short about convincingly demonstrating anything about the spatial regulation of this process.

We would like to thank the reviewer for their comments. We agree that there is much more to learn about the spatiotemporal localisation of S-layer biogenesis. We have highlighted the need for further experimental work, particularly genetic manipulation in our discussion.

Line 337

“In the future, deeper understanding of the mechanisms driving S-layer localisation would be best examined by genetic manipulation of the RsaA secretion (RsaDEF) machinery and other cell cycle proteins.”

Line 405

“For the *C. crescentus* S-layer, future studies on the localisation of RsaDEF using genetic mutants would provide further context into the mechanisms of S-layer secretion, and how this compares to other species.”

In summary (as explained above), I find that phrase in the summary “... localized S-layer

insertion is unaltered when cell division is inhibited, but depends crucially on bacterial molecules MreB and lipopolysaccharide.” is unwarranted without supporting genetic evidence.

We realise that inhibiting cell division can be performed at multiple ‘levels’, and that this can be interpreted in different ways. We have updated the text accordingly.

Line 24

“Next, light microscopy and electron cryotomography investigations of drug-treated bacteria revealed that localised S-layer insertion is retained when cell division is inhibited, but is disrupted upon dysregulation of MreB or lipopolysaccharide.”

and

Line 355

“Whether there is a higher-level coordination of S-layer biogenesis beyond local cell envelope expansion, for example through cytoskeletal localisation, is difficult to determine. Past studies on cell division and our results here suggest at least a transient co-ordination of multiple envelope components in *C. crescentus* (Fig. 9).”

Reviewer #2 (Remarks to the Author):

The authors provide extensive additional experiments that bolster the proposed model of S-layer insertion at sites of PG expansion. The cryoET imaging shows S-layer dislocations at the poles or division planes, in agreement with the proposed model that incorporation of new RsaA units are directed to sites with gaps in the S-layer. Final proof could be provided by cryoET of cells where RsaA can be selectively depleted (for example using a RsaA knockout with plasmid-encoded *rsaA* under control of *Para*), which would be expected to show gaps in the S-layer at the sites here labelled by SG-sfGFP. My support of the study is not conditional on performing this experiment, however.

We would like to thank the reviewer for their kind comments and greatly appreciate their support. We also welcome their guidance for future studies on S-layer biogenesis in *C. crescentus* and agree an RsaA depletion strain would provide important context on the gaps observed in our cryo-ET studies. We will continue to work on this in the future.